# GuidedQuant: Large Language Model Quantization via Exploiting End Loss Guidance

**Jinuk Kim** [1 2 †]  **Marwa El Halabi** [3]  **Wonpyo Park** [4]  **Clemens JS Schaefer** [4]  **Deokjae Lee** [1 2]  **Yeonhong Park** [1]
**Jae W. Lee** [1]  **Hyun Oh Song** [1 2]

## Abstract

Post-training quantization is a key technique for reducing the memory and inference latency of large language models by quantizing weights and activations without requiring retraining. However, existing methods either (1) fail to account for the varying importance of hidden features to the end loss or, when incorporating end loss, (2) neglect the critical interactions between model weights. To address these limitations, we propose GuidedQuant, a novel quantization approach that integrates gradient information from the end loss into the quantization objective while preserving cross-weight dependencies within output channels. GuidedQuant consistently boosts the performance of state-of-the-art quantization methods across weight-only scalar, weight-only vector, and weight-and-activation quantization. Additionally, we introduce a novel non-uniform scalar quantization algorithm, which is guaranteed to monotonically decrease the quantization objective value, and outperforms existing methods in this category. We release the code at https://github.com/snu-mllab/GuidedQuant.

## 1. Introduction

Large language models (LLMs) have shown remarkable capabilities across a range of tasks, from text generation to complex reasoning. However, these advancements come at the cost of substantial memory usage and inference latency. Quantization provides an effective solution to these challenges. Weight-only quantization methods quantize only the

*Table 1.* Summary of results of GuidedQuant applied to state-of-the-art PTQ methods on the Llama-2-7B model. Wiki2-4K and Wiki2-2K represent perplexity on WikiText2 dataset with context size of 4096 and 2048, respectively. W4A4KV4 indicates quantization of all weight, activation, and KV cache to 4 bits.

| Type | Method | Bits↓ | Wiki2-4K↓ |
|---|---|---|---|
| Type | Original | 16 | 5.12 |
| Weight-only Scalar | SqueezeLLM | 2.01 | 39.58 |
| | LNQ (Ours) | 2.01 | 23.31 |
| | LNQ + GQuant (Ours) | 2.01 | **8.83** |
| Weight-only Vector | QTIP | 2.00 | 6.82 |
| | QTIP + GQuant (Ours) | 2.00 | **6.11** |

| Type | Method | Bits↓ | Wiki2-2K↓ |
|---|---|---|---|
| Type | Original | 16 | 5.47 |
| Weight-and-Activation | SpinQuant | W4A4KV4 | 5.95 |
| | SpinQuant + GQuant (Ours) | W4A4KV4 | **5.89** |

model weights, reducing data transfer and thus accelerating inference in memory-bound scenarios such as small-batch inference (Gholami et al., 2024; Kim et al., 2024; Tseng et al., 2024b). On the other hand, weight-and-activation quantization methods quantize both the model weights and activations. In addition to reducing data transfer, these methods also speed up arithmetic operations, making them particularly beneficial for large-batch scenarios such as pre-filling input tokens or generating batched samples (Ashkboos et al., 2024; Liu et al., 2024). Weight-only quantization techniques have used three grid types: *uniform scalar* (Frantar et al., 2023), *non-uniform scalar* (Kim et al., 2024), and *vector* quantization (Tseng et al., 2024b; van Baalen et al., 2024), each with its own advantages (see Section 5 for details). In contrast, weight-and-activation methods typically use a uniform scalar grid, as using a non-uniform grid would require dequantization before multiplication, preventing the use of faster arithmetic operations.

Quantization benefits come at the cost of performance degradation. Quantization-Aware Training (QAT) methods rely on retraining the quantized model to mitigate this, which is prohibitively expensive at the scale of modern LLMs. In contrast, Post-Training Quantization (PTQ) methods quantize the pretrained model using a small calibration dataset or

† Work partly done during an internship at Google. [1] Department of Computer Science and Engineering, Seoul National University [2] Neural Processing Research Center [3] Samsung AI Lab, Montreal [4] Google. Correspondence to: Hyun Oh Song <hyunoh@snu.ac.kr>.

*Proceedings of the 42nd International Conference on Machine Learning*, Vancouver, Canada. PMLR 267, 2025. Copyright 2025 by the author(s).

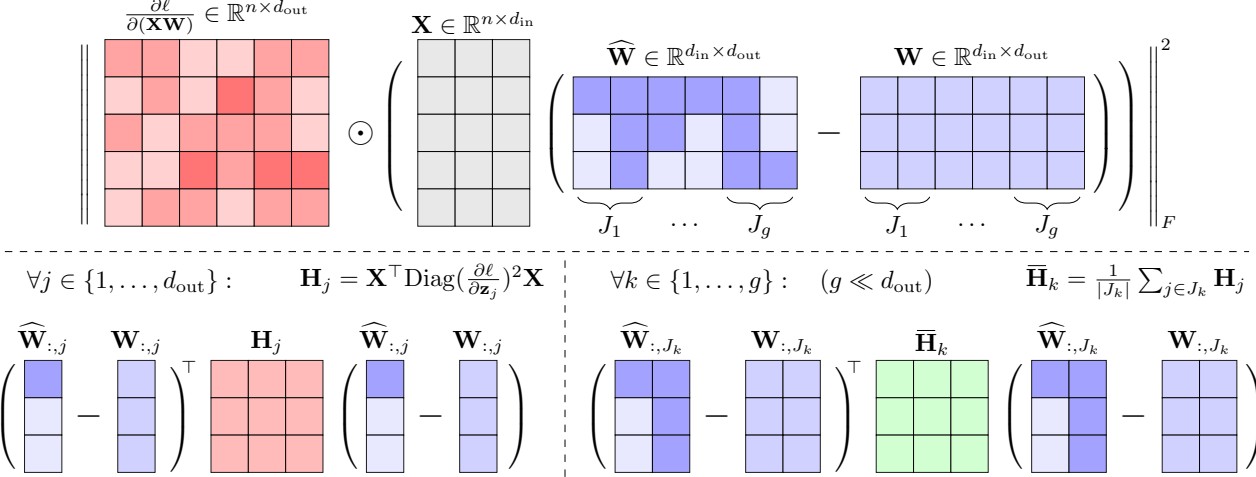

*Figure 1.* Top: The proposed GuidedQuant's layer-wise quantization objective (4). Bottom-left: Its equivalent quadratic form (6). Bottom-right: The approximated objective (7) proposed in Section 3.2. We denote the input, weight, and quantized weight matrices as $\mathbf{X} \in \mathbb{R}^{n \times d_{\text{in}}}$, $\mathbf{W} \in \mathbb{R}^{d_{\text{in}} \times d_{\text{out}}}$, and $\hat{\mathbf{W}} \in \mathbb{R}^{d_{\text{in}} \times d_{\text{out}}}$, respectively. The groups $J_1, \ldots, J_g$ form a partition of the set $\{1, \ldots, d_{\text{out}}\}$, and $\mathbf{z}_j \in \mathbb{R}^{d_{\text{out}}}$ denotes the $j$-th column of $\mathbf{Z} = \mathbf{X}\mathbf{W}$.

no data, *without* retraining the entire model. Most existing PTQ methods for LLMs rely on a surrogate objective rather than the end loss to make quantization feasible.

One common PTQ strategy, which we refer to as layer-wise output-based quantization, aims to quantize each layer by minimizing the mean squared error between the layer's original output and the quantized one (Nagel et al., 2020; Frantar et al., 2023; Egiazarian et al., 2024; Chee et al., 2024; Tseng et al., 2024a;b; Liu et al., 2024). However, this strategy treats all hidden features equally, overlooking their varying impact on the end loss.

Alternatively, methods such as Choi et al. (2017); Kim et al. (2024) leverage gradient information from the end loss to assess the impact of individual weight errors. This is done by computing the gradient of the end loss with respect to weights via a single backpropagation step on a calibration dataset. Saliency scores are then assigned to weights based on these gradients, and the model is quantized by approximately minimizing the sum of saliency-weighted weight errors. This objective corresponds to a quadratic approximation of the change in the end loss, based on its second-order Taylor expansion, where the Hessian is approximated by the *diagonal* of the empirical Fisher information matrix (Hassibi & Stork, 1992). A key limitation of this approach is that it ignores cross-weight interactions, which are crucial for overall performance.

**Contributions** In this work, we propose *GuidedQuant*, a novel PTQ approach that integrates gradient information from the end loss while preserving cross-weight dependencies within output channels. In particular, GuidedQuant

computes saliency scores for layer outputs using the gradients of the end loss with respect to these outputs. Each layer is then quantized independently by approximately minimizing the sum of saliency-weighted output errors. Unlike previous methods that assume a diagonal Hessian, this objective is equivalent to a refined quadratic approximation assuming a *block-diagonal* Hessian, again approximated by the empirical Fisher information matrix. While cross-layer and cross-output channel interactions are still ignored, dependencies within output channels are preserved, enabling a more accurate estimation of quantization's impact on the end loss.

Computing and storing the diagonal blocks of the Fisher matrix for a given layer is too expensive for modern LLMs. To address this, we partition the layer's outputs into a small number of groups and average the Fisher matrix's blocks within each group (Figure 1). Other block-diagonal Fisher matrix approximations of the Hessian have been used for pruning CNNs (Singh & Alistarh, 2020) and BERT LLMs (Kurtic et al., 2022) with arbitrary blocks along the diagonal, and for quantizing CNNs (Li et al., 2021) with diagonal blocks corresponding to the model's residual blocks (see Appendix E.11 for more details). However, our work is the first to make this approach computationally and storage-efficient at the scale of modern LLMs.

GuidedQuant can be applied as a direct plug-in to any layer-wise output-based PTQ method. We demonstrate its effectiveness by integrating it into the current state-of-the-art methods for weight-only vector quantization, QTIP (Tseng et al., 2024b), and weight-and-activation quantization, SpinQuant (Liu et al., 2024), which are both layer-wise output-

based PTQ methods. GuidedQuant consistently improves their performance (Table 1).

For weight-only scalar quantization, the current state-of-the-art methods are SqueezeLLM (Kim et al., 2024) and GPTVQ 1D (van Baalen et al., 2024). Since GPTVQ 1D is a layer-wise output-based PTQ method, GuidedQuant can be applied to it. However, GPTVQ 1D employs a suboptimal algorithm for minimizing layer-wise output errors. To address this, we introduce a novel Layer-wise Non-uniform Quantization method, *LNQ*, which minimize layer-wise output errors using an alternating minimization algorithm, where the codebook is optimized in closed-form, and assignments are optimized via a coordinate descent (CD) algorithm. LNQ outperforms GPTVQ 1D and matches or surpasses SqueezeLLM. Applying GuidedQuant to LNQ further improves its performance, achieving state-of-the-art results (Table 1).

## 2. Preliminaries

Consider a neural network with $L$ linear layers, trained with a loss function $\ell$ and a calibration data of size $n$. We denote the loss computed on the $i$-th data point as $\ell_i$. Let $\mathbf{W}^{(l)} \in \mathbb{R}^{d_{in}^{(l)} \times d_{out}^{(l)}}$ be the weight matrix of the $l$-th linear layer, where each column vector $\mathbf{w}_j^{(l)} \in \mathbb{R}^{d_{in}^{(l)}}$ corresponds to an output channel. We denote its quantized approximation as $\widehat{\mathbf{W}}^{(l)}$. The input and output feature maps of this layer are $\mathbf{X}^{(l)} \in \mathbb{R}^{n \times d_{in}^{(l)}}$ and $\mathbf{Z}^{(l)} \in \mathbb{R}^{n \times d_{out}^{(l)}}$, respectively. The output of the linear layer is computed as $\mathbf{Z}^{(l)} = \mathbf{X}^{(l)}\mathbf{W}^{(l)}$, and the output after quantization as $\widehat{\mathbf{Z}}^{(l)} = \mathbf{X}^{(l)}\widehat{\mathbf{W}}^{(l)}$. Let $\mathbf{w} = [\text{vec}(\mathbf{W}^{(1)})^\top, \cdots, \text{vec}(\mathbf{W}^{(L)})^\top]^\top$ and $\widehat{\mathbf{w}} = [\text{vec}(\widehat{\mathbf{W}}^{(1)})^\top, \cdots, \text{vec}(\widehat{\mathbf{W}}^{(L)})^\top]^\top$ be the vectors of weights in all $L$ layers before and after quantization, where $\text{vec}(\mathbf{W}^\ell)$ corresponds to stacking the columns of $\mathbf{W}^\ell$.

Most existing PTQ methods for LLMs are layer-wise output-based quantization methods, which quantize each layer by approximately minimizing the objective

$$\|\mathbf{X}^{(l)}\mathbf{W}^{(l)} - \mathbf{X}^{(l)}\widehat{\mathbf{W}}^{(l)}\|_F^2 = \sum_{i=1}^{n}\sum_{j=1}^{d_{out}^{(l)}}\left(Z_{ij}^{(l)} - \widehat{Z}_{ij}^{(l)}\right)^2 \quad (1)$$

ignoring the varying impact of outputs on the end loss $\ell$. Existing methods employ various heuristics to minimize this objective, such as AdaRound (Nagel et al., 2020), CD methods (Nair & Suggala, 2024; Behdin et al., 2023; Egiazarian et al., 2024; Chee et al., 2024), OBQ (Frantar & Alistarh, 2022), GPTQ[1] (Frantar et al., 2023), GPTVQ (van Baalen et al., 2024), and AQLM (Egiazarian et al., 2024).

A more accurate proxy objective, first introduced in early pruning methods (LeCun et al., 1989; Hassibi & Stork,

---

[1]Also referred to as OPTQ.

1992), is the following quadratic approximation of the change in the end loss

$$\ell(\widehat{\mathbf{w}}) - \ell(\mathbf{w}) \approx \tfrac{1}{2}(\widehat{\mathbf{w}} - \mathbf{w})^\top \nabla^2 \ell(\mathbf{w})(\widehat{\mathbf{w}} - \mathbf{w}). \quad (2)$$

This approximation is derived from the second-order Taylor approximation of $\ell$, assuming that the trained model has converged and thus the gradient is close to zero. Since computing the Hessian is infeasible even for small models, a popular approach first proposed in Hassibi & Stork (1992) approximates the Hessian by the empirical Fisher information matrix $\mathbf{F} = \frac{1}{n}\sum_{i=1}^{n}\nabla\ell_i(\mathbf{w})\nabla\ell_i(\mathbf{w})^\top$, which yields the following quadratic approximation, $(\widehat{\mathbf{w}} - \mathbf{w})^\top \mathbf{F}(\widehat{\mathbf{w}} - \mathbf{w})$.

SqueezeLLM (Kim et al., 2024) is a weight-only non-uniform scalar PTQ method for LLMs which uses this quadratic approximation, but further approximates the Fisher information matrix by its diagonal $\text{diag}(\mathbf{F})$, ignoring off-diagonal entries. The resulting objective is given by

$$(\widehat{\mathbf{w}} - \mathbf{w})^\top \text{diag}(\mathbf{F})(\widehat{\mathbf{w}} - \mathbf{w}) = \sum_k F_{kk}(\widehat{w}_k - w_k)^2. \quad (3)$$

For non-uniform scalar quantization, minimizing this objective corresponds to solving a weighted $k$-means problem in 1D, which can be solved exactly using a dynamic programming algorithm (Grønlund et al., 2017). SqueezeLLM instead employs Lloyd's algorithm with $k$-means++ initialization (Lloyd, 1982; Arthur & Vassilvitskii, 2007), which is only guaranteed to achieve a $\Theta(\log k)$ approximation in expectation, where $k$ is the number of clusters, but is faster in practice (Hyun, 2024). However, the diagonal approximation is highly inaccurate, as both the Hessian matrix and its Fisher approximation are usually strongly non-diagonal, as observed in prior work for small CNNs (Hassibi & Stork, 1992; Singh & Alistarh, 2020). We also confirm this observation for the Fisher matrix of Llama-2-7B in Figures 3 and 4.

For additional related work, see Appendix A.

## 3. GuidedQuant

In this section, we introduce our PTQ approach *GuidedQuant*. We first propose a layer-wise quantization objective that more accurately approximates the impact of quantization on the final loss compared to surrogate objectives used in existing PTQ methods. We then present a simplified version of this objective, making it computationally and memory efficient for LLMs with up to 70B parameters.

### 3.1. Objective

As discussed earlier, most existing PTQ methods treat all output features as equally important, by employing the surrogate objective in Eq. (1). In contrast, we propose to modify this objective to account for the varying impact of each output feature on the final loss.

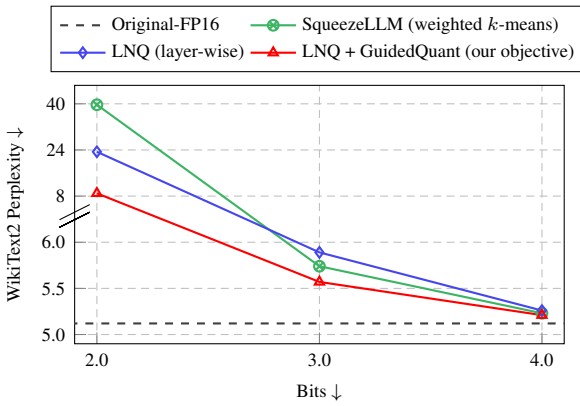

*Figure 2.* Non-uniform scalar quantization results on Llama-2-7B with different objectives: *layer-wise* output error objective (1) used in LNQ (Algorithm 2), *weighted k-means* objective (3) used in SqueezeLLM, and our approximated *GuidedQuant* objective (7) used in LNQ combined with GuidedQuant. We report perplexity on WikiText2 with a context size of 4096. Results are from Table 3.

To that end, we approximate the change in the end loss $\ell$ resulting from the output feature $Z_{ij}^{(l)}$ changing to $\widehat{Z}_{ij}^{(l)}$ after quantization, using a first-order Taylor expansion, assuming independence of output features:

$$\ell(\widehat{Z}_{ij}^{(l)}) - \ell(Z_{ij}^{(l)}) \approx \frac{\partial \ell}{\partial Z_{ij}^{(l)}}(\widehat{Z}_{ij}^{(l)} - Z_{ij}^{(l)}).$$

Accordingly, we propose to scale each output error by the gradient of the end loss with respect to that output, leading to the following layer-wise objective:

$$\left\| \frac{\partial \ell}{\partial \mathbf{Z}^{(l)}} \odot (\mathbf{X}^{(l)} \mathbf{W}^{(l)} - \mathbf{X}^{(l)} \widehat{\mathbf{W}}^{(l)}) \right\|_F^2$$
$$= \sum_{i=1}^{n} \sum_{j=1}^{d_{\text{out}}^{(l)}} \left( \frac{\partial \ell}{\partial Z_{ij}^{(l)}}(Z_{ij}^{(l)} - \widehat{Z}_{ij}^{(l)}) \right)^2, \qquad (4)$$

where $\odot$ denotes the element-wise multiplication. This criterion was previously proposed in Molchanov et al. (2019) for pruning neurons and filters in vision models, where pruning the $j$th neuron in layer $l$ corresponds to setting $\widehat{\mathbf{w}}_j^{(l)} = 0$.

We note that the objective in Eq. (4) can be viewed as a simplification of the second-order Taylor approximation of the change in the end loss given in Eq. (2), where the Hessian is approximated by the empirical Fisher information matrix, and where interactions between weights belonging to different layers or output channels of the same layer are ignored. In other words, we adopt a *block-diagonal* approximation of the Fisher matrix $\mathbf{F}$ where we only keep the $d_{\text{in}}^{(l)} \times d_{\text{in}}^{(l)}$ blocks $\mathbf{F}_j^{(l)} = \frac{1}{n} \sum_{i=1}^{n} (\frac{\partial \ell_i}{\partial \mathbf{w}_j^{(l)}})(\frac{\partial \ell_i}{\partial \mathbf{w}_j^{(l)}})^\top$ corresponding to interactions within each output channel $j$ of every layer $l$, and ignore all off-block entries.

*Remark* 3.1. The sum of the layer-wise objective in Eq. (4) over all layers is equal to the following quadratic approximation of the change in the end loss

$$n \sum_{l=1}^{L} \sum_{j=1}^{d_{\text{out}}^{(l)}} (\mathbf{w}_j^{(l)} - \widehat{\mathbf{w}}_j^{(l)})^\top \mathbf{F}_j^{(l)} (\mathbf{w}_j^{(l)} - \widehat{\mathbf{w}}_j^{(l)}). \qquad (5)$$

The proof of Remark 3.1 follows from the chain rule, and is given in Appendix B. A similar observation was made in Molchanov et al. (2019).

Assuming that the quantization grid used is separable over layers, which is typically the case, minimizing the objective in Eq. (5) is equivalent to independently minimizing

$$\sum_{j=1}^{d_{\text{out}}^{(l)}} (\mathbf{w}_j^{(l)} - \widehat{\mathbf{w}}_j^{(l)})^\top \mathbf{H}_j^{(l)} (\mathbf{w}_j^{(l)} - \widehat{\mathbf{w}}_j^{(l)}), \qquad (6)$$

for every layer, where $\mathbf{H}_j^{(l)} = n\mathbf{F}_j^{(l)}$, or equivalently the layer-wise objective in Eq. (4).

Thus our proposed objective is a more accurate approximation of the change in the end loss than the layer-wise output error objective (1), which assumes $\frac{\partial \ell}{\partial \mathbf{Z}^{(l)}} \propto \mathbf{I}$, as well as the weighted $k$-means objective (3) used in SqueezeLLM, which ignores all off-diagonal entries in the Fisher matrix including those within the blocks $\mathbf{F}_j^{(l)}$. As a result, our approach achieves better performance, even with the additional approximation discussed in Section 3.2, as highlighted in Figure 2 for non-uniform scalar quantization, and later across other formats in Section 5.

In Figures 3 and 4, we visualize a submatrix of the Fisher information matrix corresponding to the first two output channels in the linear layers of the first Transformer block of Llama-2-7B. The visualization confirms that the Fisher matrix exhibits strong off-diagonal values and a prominent block-diagonal structure, with blocks corresponding to $\mathbf{F}_j^{(l)}$ for the two output channels $j \in \{1, 2\}$.

### 3.2. Averaging Approximation

The layer-wise output error objective (1) can be written as

$$\sum_{j=1}^{d_{\text{out}}^{(l)}} \left( \mathbf{w}_j^{(l)} - \widehat{\mathbf{w}}_j^{(l)} \right)^\top \mathbf{H}^{(l)} \left( \mathbf{w}_j^{(l)} - \widehat{\mathbf{w}}_j^{(l)} \right),$$

where $\mathbf{H}^{(l)} = \mathbf{X}^{(l)\top} \mathbf{X}^{(l)} \in \mathbb{R}^{d_{\text{in}}^{(l)} \times d_{\text{in}}^{(l)}}$. Most existing heuristics for optimizing this objective, such as GPTQ (Frantar et al., 2023) and CD (Nair & Suggala, 2024; Behdin et al., 2023), require access only to $\mathbf{H}^{(l)}$ and not $\mathbf{X}^{(l)}$. Thus, the Hessian matrix $\mathbf{H}^{(l)}$ is typically precomputed, which reduces the peak memory usage during optimization, since $\mathbf{X}^{(l)} \in \mathbb{R}^{n \times d_{\text{in}}^{(l)}}$ is much larger than $\mathbf{H}^{(l)}$, given

that $d_{\text{in}}^{(l)} \ll n$. Additionally, the precomputed Hessian can be reused across multiple quantization configurations and bit-widths, amortizing the cost of its computation.

Our proposed objective (6) can be seamlessly integrated into any layer-wise output based quantization method by replacing $\mathbf{H}^{(l)}$ by $\mathbf{H}_j^{(l)} = n\mathbf{F}_j^{(l)}$ for each output channel $j$. However, precomputing and storing $\mathbf{H}_j^{(l)}$ for all $j$ incurs a memory cost of $\Theta((d_{\text{in}}^{(l)})^2 d_{\text{out}}^{(l)})$ and a time complexity of $\Theta(n(d_{\text{in}}^{(l)})^2 d_{\text{out}}^{(l)})$ per layer $l$. This is infeasible at the scale of modern LLMs, where both $d_{\text{in}}^{(l)}$ and $d_{\text{out}}^{(l)}$ exceed $10^3$, and $n$ is much larger than both.

To address this challenge, we partition the output channels of each layer into $g$ distinct groups ($g \ll d_{\text{out}}^{(l)}$) and replace the individual Hessian matrices $\mathbf{H}_j^{(l)}$ within each group $k$ by a shared matrix $\overline{\mathbf{H}}_k^{(l)}$, obtained by averaging $\mathbf{H}_j^{(l)}$ within the group. Formally, let $J_1^{(l)}, \dots, J_g^{(l)}$ be a partition of the set $\{1, \dots, d_{\text{out}}^{(l)}\}$. For each group $k = 1, \dots, g$, we define $\overline{\mathbf{H}}_k^{(l)} = \frac{1}{|J_k^{(l)}|} \sum_{j \in J_k^{(l)}} \mathbf{H}_j^{(l)}$. The resulting layer-wise objective then becomes

$$\sum_{k=1}^{g} \sum_{j \in J_k^{(l)}} \left( \mathbf{w}_j^{(l)} - \widehat{\mathbf{w}}_j^{(l)} \right)^\top \overline{\mathbf{H}}_k^{(l)} \left( \mathbf{w}_j^{(l)} - \widehat{\mathbf{w}}_j^{(l)} \right). \quad (7)$$

Note that by the chain rule, we can write

$$\mathbf{H}_j^{(l)} = \mathbf{X}^{(l)\top} \text{Diag} \left( \frac{\partial \ell}{\partial \mathbf{z}_j^{(l)}} \right)^2 \mathbf{X}^{(l)},$$

where $\text{Diag}(\frac{\partial \ell}{\partial \mathbf{z}_j^{(l)}})^2$ is the diagonal matrix whose diagonal entries are the element-wise square of the gradient of $\ell$ with respect to the $j$th column $\mathbf{z}_j^{(l)}$ of $\mathbf{Z}^{(l)}$. We can thus compute $\overline{\mathbf{H}}_k^{(l)}$ by averaging the squared gradients:

$$\overline{\mathbf{H}}_k^{(l)} = \mathbf{X}^{(l)\top} \text{Diag} \left( \frac{1}{|J_k|} \sum_{j \in J_k} \left( \frac{\partial \ell}{\partial \mathbf{z}_j^{(l)}} \right)^2 \right) \mathbf{X}^{(l)}.$$

This averaging approximation reduces the number of $d_{\text{in}}^{(l)} \times d_{\text{in}}^{(l)}$ Hessian matrices that need to be computed for each layer $l$ from $d_{\text{out}}^{(l)}$ to $g$ (Figure 1). Computing and storing $\overline{\mathbf{H}}_k^{(l)}$ for all $k$ requires a significantly lower memory cost of $\Theta((d_{\text{in}}^{(l)})^2 g)$ and time complexity of $\Theta(n(d_{\text{in}}^{(l)})^2 g)$ per layer $l$ (assuming the squared gradients averages are already computed), making the method scalable. To partition the output channels, we use a simple strategy that groups every $d_{\text{out}}^{(l)}/g$ consecutive channels into a single group. This simple approach works well in practice, though more sophisticated clustering algorithms may yield additional benefits.

**Algorithm 1** GuidedQuant

**input** Layer-wise quantization algorithm $\mathcal{Q}$, number of groups $g$, number of linear layers $L$

1: $J_k^{(l)} \leftarrow \{ \frac{d_{\text{out}}^{(l)}}{g}(k-1) + 1, \dots, \frac{d_{\text{out}}^{(l)}}{g} k \}, \forall l \in [L], k \in [g]$
2: $\mathbf{s}_k^{(l)} \leftarrow \frac{1}{|J_k|} \sum_{j \in J_k} (\frac{\partial \ell}{\partial \mathbf{z}_j^{(l)}})^2, \forall l \in [L], k \in [g]$
3: **for all** $l \in [L], k \in [g]$ **do**
4: $\quad \overline{\mathbf{H}}_k^{(l)} \leftarrow \mathbf{X}^{(l)\top} \text{Diag}(\mathbf{s}_k^{(l)}) \mathbf{X}^{(l)}$
5: $\quad \widehat{\mathbf{W}}^{(l)} \left[ :, J_k^{(l)} \right] \leftarrow \mathcal{Q} \left( \overline{\mathbf{H}}_k^{(l)}, \mathbf{W}^{(l)} \left[ :, J_k^{(l)} \right] \right)$
6: **end for**
**output** $\widehat{\mathbf{W}}^{(1)}, \dots, \widehat{\mathbf{W}}^{(L)}$.

In our implementation, we scale the gradients by a large constant (we used $10^3$ in all experiments) while computing the averaged Hessians $\overline{\mathbf{H}}_k$ to prevent underflow.

GuidedQuant quantizes each layer independently by approximately minimizing the layer-wise objective in Eq. (7). A complete overview of GuidedQuant is provided in Algorithm 1. As discussed, the layer-wise quantization algorithm $\mathcal{Q}$ can be any layer-wise output based quantization method. The gradient computation (Line 2) requires a single backpropagation step on the calibration dataset. During this step, we only store the averaged squared gradients $\mathbf{s}_k^{(l)}$, which requires $O(ngL)$ storage.

The total memory cost of GuidedQuant (without the backpropagation step) is then $O(Lg(d_{\text{in}}^2 + n))$, and its total time complexity is $O\left(Lg(nd_{\text{in}}^2 + T_{\mathcal{Q}}(d_{\text{in}}, d_{\text{out}}/g))\right)$, where $d_{\text{in}}, d_{\text{out}}$ are the largest input and output channel dimensions across all $L$ layers and $T_{\mathcal{Q}}(d_1, d_2)$ is the time complexity of quantizing a $d_1 \times d_2$-weight matrix using $\mathcal{Q}$. Each step in the for loop (Lines 3-6) can be done in parallel for all groups and layers. As previously discussed, the Hessian matrices $\overline{\mathbf{H}}_k$'s only need to be computed once, and can be reused for different quantization configurations and bit-widths.

## 4. Layer-wise Non-uniform Quantization

The choice of the layer-wise output based quantization method $\mathcal{Q}$ in GuidedQuant is critical to its overall performance. For weight-only non-uniform scalar quantization, the current state-of-the-art layer-wise output based quantization method is the 1D variant of GPTVQ (van Baalen et al., 2024), which alternates between optimizing the codebook via gradient descent and the assignments via GPTQ algorithm (Frantar et al., 2023). However, both of these steps can be improved. Given fixed assignments, the codebook admits an optimal closed form solution. Also, for optimizing assignments, recent works have demonstrated that coordinate descent (CD) methods outperform GPTQ in uniform weight-only quantization (Behdin et al., 2023; Nair & Suggala, 2024). In this section, we introduce Layer-wise

---

**Algorithm 2** LNQ

---

**input** Hessian of the objective $\mathbf{H} \in \mathbb{R}^{d_{\text{in}} \times d_{\text{in}}}$, input weight $\mathbf{W} \in \mathbb{R}^{d_{\text{in}} \times d_{\text{out}}}$, initial assignment $\mathbf{P}^{(j)} \in \mathbb{R}^{d_{\text{in}} \times m}$ for each output channel $j$.

1:   $\mathbf{H} = \mathbf{L}\mathbf{L}^{\top}$             {*Cholesky decomposition*}
2:   **for** $j \in \{1, \ldots, d_{\text{out}}\}$ **do**
3:     **for** $t = 1$ **to** $T$ **do**
4:       $\mathbf{c}^{(j)} \leftarrow \left(\mathbf{P}^{(j)\top}\mathbf{L}\mathbf{L}^{\top}\mathbf{P}^{(j)}\right)^{-1}\mathbf{P}^{(j)\top}\mathbf{L}\mathbf{L}^{\top}\mathbf{w}_j$
5:       $\hat{\mathbf{w}}_j \leftarrow \mathbf{P}^{(j)}\mathbf{c}^{(j)}$
6:       **for** $k = 1$ **to** $K$ **do**
7:         **for** $i = 1$ **to** $d_{\text{in}}$ **do**
8:           $c_{q^*}^{(j)} \leftarrow \underset{\widehat{W}_{ij} \in \{c_1^{(j)}, \ldots, c_m^{(j)}\}}{\text{argmin}} \; (\hat{\mathbf{w}}_j - \mathbf{w}_j)^{\top}\mathbf{H}(\hat{\mathbf{w}}_j - \mathbf{w}_j)$
9:           $\widehat{W}_{ij} \leftarrow c_{q^*}^{(j)}$
10:          $\forall q \in \{1, \ldots, m\} : P_{iq}^{(j)} = \begin{cases} 1 & \text{if } q = q^*, \\ 0 & \text{otherwise.} \end{cases}$
11:         **end for**
12:       **end for**
13:     **end for**
14:     $\mathbf{c}^{(j)} \leftarrow \left(\mathbf{P}^{(j)\top}\mathbf{L}\mathbf{L}^{\top}\mathbf{P}^{(j)}\right)^{-1}\mathbf{P}^{(j)\top}\mathbf{L}\mathbf{L}^{\top}\mathbf{w}_j$
15: **end for**
**output**   $\widehat{\mathbf{W}} = [\mathbf{P}^{(1)}\mathbf{c}^{(1)}, \ldots, \mathbf{P}^{(d_{\text{out}})}\mathbf{c}^{(d_{\text{out}})}]$.

---

Non-uniform Quantization (LNQ), an alternating minimization algorithm which leverages the closed form solution for the codebook and employs CD to optimize the assignments. We then discuss its theoretical guarantees, as well as the memory cost and computational complexity under our efficient implementation.

### 4.1. Optimization Problem

We omit the layer index $l$ for notational simplicity throughout this section. Following prior work, we assign to each output channel a separate codebook, though LNQ can be easily adapted to finer-granularity grouping. Non-uniform scalar quantization maps each scalar weight in the column $\mathbf{w}_j \in \mathbb{R}^{d_{\text{in}}}$ to one of $m = 2^b$ real values $\{c_1^{(j)}, \ldots, c_m^{(j)}\}$, where $b \in \mathbb{N}$ is the target bit-width. The quantized weights $\hat{\mathbf{w}}_j$ can then be expressed as $\hat{\mathbf{w}}_j = \mathbf{P}^{(j)}\mathbf{c}^{(j)}$, where $\mathbf{c}^{(j)} \in \mathbb{R}^m$ is the vector containing the codebook values $\{c_1^{(j)}, \ldots, c_m^{(j)}\}$, and $\mathbf{P}^{(j)} \in \{0, 1\}^{d_{\text{in}} \times m}$ is the assignment matrix such that $P_{iq}^{(j)} = 1$ if $W_{ij}$ is assigned to $c_q^{(j)}$, and $P_{iq}^{(j)} = 0$ otherwise.

The optimization problem for layer-wise output-based non-uniform scalar quantization can then be written as follows:

$$\underset{\substack{\mathbf{P}^{(j)} \in \{0,1\}^{d_{\text{in}} \times m} \\ \mathbf{c}^{(j)} \in \mathbb{R}^m}}{\text{minimize}} \quad \sum_{j=1}^{d_{\text{out}}} \|\mathbf{X}\mathbf{w}_j - \mathbf{X}\mathbf{P}^{(j)}\mathbf{c}^{(j)}\|_2^2$$
$$\text{subject to} \quad \mathbf{P}^{(j)}\mathbf{1}_m = \mathbf{1}_{d_{\text{in}}}, \tag{8}$$

where $\mathbf{1}$ is the vector of all ones. Note that the optimization for each column $j$ is independent of other columns, and can be done in parallel.

### 4.2. LNQ Algorithm

We propose LNQ, an alternating minimization algorithm, which iteratively updates the codebook $\mathbf{c}^{(j)}$ and assignment matrix $\mathbf{P}^{(j)}$ for each $j$, optimizing one while keeping the other fixed. Alternating minimization is a common strategy used by most non-uniform quantization methods, including SqueezeLLM and GPTVQ. LNQ quantizes each layer independently. We present an overview of LNQ, applied to one layer with weights $\mathbf{W} \in \mathbb{R}^{d_{\text{in}} \times d_{\text{out}}}$ in Algorithm 2.

Given fixed assignment matrices $\mathbf{P}^{(j)}$, Problem (8) reduces to a standard least-squares problem, which admits a closed-form optimal solution $\mathbf{c}^{(j)*} = (\mathbf{X}\mathbf{P}^{(j)})^{\dagger}\mathbf{X}\mathbf{w}_j$, where $\dagger$ denotes the Moore–Penrose pseudoinverse. We assume that the matrix $\mathbf{P}^{(j)\top}\mathbf{H}\mathbf{P}^{(j)}$ is invertible, where recall that $\mathbf{H} = \mathbf{X}^{\top}\mathbf{X}$. Under this assumption, the closed-form solution is:

$$\mathbf{c}^{(j)*} = \left(\mathbf{P}^{(j)\top}\mathbf{H}\mathbf{P}^{(j)}\right)^{-1}\mathbf{P}^{(j)\top}\mathbf{H}\mathbf{w}_j. \tag{9}$$

In practice, $\mathbf{P}^{(j)\top}\mathbf{H}\mathbf{P}^{(j)}$ is not always invertible, even when $\mathbf{H}$ is invertible (for example if no weight is assigned to a given codebook value $c_q^{(j)}$). To address this, we add a small constant $\lambda = 10^{-7}$ to the diagonal of the matrix, as commonly done in prior work (Frantar & Alistarh, 2022; Frantar et al., 2023; van Baalen et al., 2024). In our implementation, we use `torch.linalg.lstsq` function to compute the least squares solution in Eq. (9), which takes $\mathbf{X}\mathbf{P}^{(j)}$ and $\mathbf{X}\mathbf{w}_j$ as inputs. However, since $\mathbf{X}$ is not explicitly stored, we compute the Cholesky decomposition of $\mathbf{H} = \mathbf{X}^{\top}\mathbf{X}$, denoted as $\mathbf{H} = \mathbf{L}\mathbf{L}^{\top}$, and instead provide $\mathbf{L}^{\top}\mathbf{P}^{(j)}$ and $\mathbf{L}\mathbf{w}_j$ to the solver. Because Cholesky decomposition requires $\mathbf{H}$ to be positive definite, we ensure this by adding a small constant to the diagonal of $\mathbf{H}$.

For fixed codebooks $\mathbf{c}^{(j)}$, Problem (8) can be equivalently written as

$$\underset{\hat{\mathbf{w}}_j \in \{c_1^{(j)}, \ldots, c_m^{(j)}\}^{d_{\text{in}}}}{\text{minimize}} \sum_{j=1}^{d_{\text{out}}} (\hat{\mathbf{w}}_j - \mathbf{w}_j)^{\top}\mathbf{H}(\hat{\mathbf{w}}_j - \mathbf{w}_j). \tag{10}$$

Even in the special case of uniform codebook, this problem corresponds to a closest vector problem with box constraints, which is NP-Hard to approximate within any constant factor approximation for $m \geq 2$ (Arora et al., 1997, Theorem 1). Existing heuristics for solving it include OBQ (Frantar & Alistarh, 2022) which does not scale to LLMs with billions of parameters; its faster variant GPTQ (Frantar et al., 2023); LDLQ (Chee et al., 2024), which is a more efficient implementation of GPTQ; greedy CD (Nair & Suggala, 2024); and cyclic CD (Behdin et al., 2023; Chee et al., 2024;

Egiazarian et al., 2024). Recent works show that both greedy CD (Nair & Suggala, 2024) and cyclic CD (Behdin et al., 2023) outperform GPTQ on this problem when using a uniform grid. We thus adopt the cyclic CD algorithm, since it performs similarly to the greedy variant while being significantly less expensive (Nair & Suggala, 2024, Appendix D). In Appendix E.6, we present an ablation study that further support this choice, showing that cyclic CD matches or outperforms GPTQ when used within LNQ for non-uniform scalar quantization.

Cyclic CD is an iterative algorithm which iterates over coordinates in a fixed order, minimizing at each iteration the objective with respect to one coordinate, while keeping all others fixed. The minimization for each coordinate (Line 8 in Algorithm 2) has a closed form solution, as shown in Behdin et al. (2023, Lemma 1) and Chee et al. (2024, Section B.2):

$$\text{Round}_j \left( W_{i,j} - \frac{\mathbf{H}_{i,[d_{\text{in}}]\setminus i}}{H_{i,i}} (\widehat{\mathbf{W}}_{[d_{\text{in}}]\setminus i,j} - \mathbf{W}_{[d_{\text{in}}]\setminus i,j}) \right), \tag{11}$$

where $\text{Round}_j(\cdot)$ denotes rounding to the nearest point in the grid $\{c_1^{(j)}, \ldots, c_m^{(j)}\}$.

CD is a descent method when initialized with a feasible solution $\widehat{\mathbf{w}}_j \in \{c_1^{(j)}, \ldots, c_m^{(j)}\}^{d_{\text{in}}}$, i.e., it monotonically decreases the objective function value. It can be used as a standalone solver for problem (10) initialized with the original weights $\mathbf{W}$, as in Nair & Suggala (2024); Behdin et al. (2023), or to refine the output of another quantization method, as done in the uniform quantization method QuIP, which runs CD after LDLQ (Chee et al., 2024).

In LNQ, at each iteration, we initialize CD with the quantized weights corresponding to the current assignment and codebook $\hat{\mathbf{w}}_j = \mathbf{P}^{(j)}\mathbf{c}^{(j)}$ for each $j$. For the first iteration, any feasible assignment matrix can be used. In our experiments, we initialize with the assignments from SqueezeLLM. Since the codebooks are updated optimally and CD acts as descent method with feasible initialization, it follows that LNQ itself is a descent method and it converges. Refer to Appendix B for the proof.

**Proposition 4.1.** *For any* $j \in [d_{\text{out}}]$, *let* $f_j(\mathbf{c}, \mathbf{P}) = \|\mathbf{X}\mathbf{w}_j - \mathbf{X}\mathbf{P}\mathbf{c}\|_2^2$, *and let* $\mathbf{c}_t^{(j)}$ *and* $\mathbf{P}_t^{(j)}$ *denote* $\mathbf{c}^{(j)}$ *and* $\mathbf{P}^{(j)}$ *at the* $t$-*th iteration of LNQ. Then,* $f_j(\mathbf{c}_t^{(j)}, \mathbf{P}_t^{(j)}) \geq f_j(\mathbf{c}_{t+1}^{(j)}, \mathbf{P}_t^{(j)}) \geq f_j(\mathbf{c}_{t+1}^{(j)}, \mathbf{P}_{t+1}^{(j)})$ *for all* $t$, *and the sequence* $\{f_j(\mathbf{c}_t^{(j)}, \mathbf{P}_t^{(j)})\}_{t \geq 1}$ *converges.*

Since LNQ is a layer-wise output based method, GuidedQuant can be easily applied to it. In Section 5.1, we demonstrate the efficacy of LNQ both as a standalone approach and in combination with the GuidedQuant objective.

**Time Complexity.** Computing the Cholesky decomposition of $\mathbf{H}$ (Line 1) requires $O(d_{\text{in}}^3)$, optimizing the code-

*Table 2.* End-to-end inference throughput of Llama-2 models on RTX 4090 GPU. OOM indicates an Out-of-Memory error, meaning the GPU lacks memory to run model inference. See Appendix D.1 for experimental setup details.

| | Llama-2-7B | | Llama-2-13B | | Llama-2-70B | |
|---|---|---|---|---|---|---|
| Type | Bits↓ | Tok/s↑ | Bits↓ | Tok/s↑ | Bits↓ | Tok/s↑ |
| Original | 16 | 67 | 16 | OOM | 16 | OOM |
| Uniform scalar | 2.00 | 334 | 2.00 | 200 | 2.00 | 47 |
| Non-uniform scalar | 2.01 | 347 | 2.01 | 203 | 2.01 | 47 |
| Vector | 2.00 | 200 | 2.00 | 121 | 2.00 | 38 |
| Uniform scalar | 3.00 | 260 | 3.00 | 150 | 3.00 | OOM |
| Non-uniform scalar | 3.03 | 264 | 3.02 | 148 | 3.01 | OOM |
| Vector | 3.00 | 176 | 3.00 | 103 | 3.00 | OOM |
| Uniform scalar | 4.00 | 214 | 4.00 | 121 | 4.00 | OOM |
| Non-uniform scalar | 4.05 | 209 | 4.04 | 116 | 4.03 | OOM |
| Vector | 4.00 | 151 | 4.00 | 89 | 4.00 | OOM |

book (Line 4 and 14) requires $O(d_{\text{in}}^2 m)$, and optimizing the codes (Lines 6-12) requires $O(d_{\text{in}}^2 K)$ time complexity. The total time complexity of LNQ algorithm is $O(d_{\text{in}}^3 + d_{\text{in}}^2 d_{\text{out}} T(m + K))$. Here, $T$ and $K$ denotes the number of iterations for alternating optimization and the number of cycles in coordinate descent, respectively. We provide a detailed analysis of the time complexity in Appendix C.2. We discuss in Appendix C.3 how to significantly speedup the implementation of CD on GPU, using precomputation and lazy batch-updates. Precomputation is also used in Behdin et al. (2023); Chee et al. (2024), while lazy batch-updates is only used in Chee et al. (2024) (though not discussed in the paper). These tricks do no change the theoretical time complexity of CD, but they yield up to $3\times$ speedups in practice.

## 5. Experiments

In this section, we demonstrate the versatility and effectiveness of our method across various quantization schemes. We first explore different quantization scenarios and identify the formats best suited to each setting, ultimately focusing on three main approaches: weight-only scalar, weight-only vector, and weight-and-activation quantization. By integrating the GuidedQuant objective into existing methods, our results consistently achieve state-of-the-art PTQ performance. Refer to Appendix D.2 for details on how we incorporate GuidedQuant objective into existing methods. Additional experiments and details, including the overall cost of our method, the effect of the number of groups $g$, and the end-to-end fine-tuning results, are provided in Appendix E.

### 5.1. Weight-only Quantization

**Experimental Setup.** Weight-only quantization primarily accelerates inference latency in low-batch scenarios, where memory bandwidth constitutes the main bottleneck

*Table 3.* Weight-only scalar post-training quantization results *without fine-tuning* with end-to-end loss. Wiki2 and C4 denotes perplexity on WikiText2 and C4, respectively. The perplexity is measured with the context size of 4096.

| Method | Llama-2-7B | | | Llama-2-13B | | | Llama-2-70B | | |
|---|---|---|---|---|---|---|---|---|---|
| | Bits↓ | Wiki2↓ | C4↓ | Bits↓ | Wiki2↓ | C4↓ | Bits↓ | Wiki2↓ | C4↓ |
| Original | 16 | 5.12 | 6.63 | 16 | 4.57 | 6.05 | 16 | 3.12 | 4.97 |
| QuIP | – | – | – | 2.00 | 13.48 | 16.16 | 2.01 | 5.90 | 8.17 |
| SqueezeLLM | 2.01 | 39.58 | 44.05 | 2.01 | 16.24 | 19.20 | 2.01 | 9.17 | 13.03 |
| GPTVQ 1D | 2.03 | 51.87 | 47.33 | 2.03 | 9.53 | 12.62 | 2.03 | 6.03 | 8.44 |
| LNQ (Ours) | 2.01 | 23.31 | 26.71 | 2.01 | 8.78 | 11.80 | 2.01 | 5.23 | 7.31 |
| LNQ + GuidedQuant (Ours) | 2.01 | **8.83** | **11.15** | 2.01 | **7.26** | **9.17** | 2.01 | **5.04** | **7.04** |
| GPTQ | 3.00 | 8.06 | 10.61 | 3.00 | 5.85 | 7.86 | 3.00 | 4.40 | 6.26 |
| QuIP | – | – | – | 3.00 | 5.12 | 6.79 | 3.01 | 3.87 | 5.67 |
| SqueezeLLM | 3.03 | 5.74 | 7.44 | 3.02 | 4.99 | 6.60 | 3.01 | 3.53 | 5.31 |
| GPTVQ 1D | 3.03 | 6.17 | 8.02 | 3.03 | 5.13 | 6.76 | 3.03 | 3.55 | 5.35 |
| LNQ (Ours) | 3.03 | 5.89 | 7.74 | 3.02 | 5.02 | 6.68 | 3.01 | 3.50 | 5.31 |
| LNQ + GuidedQuant (Ours) | 3.03 | **5.57** | **7.22** | 3.02 | **4.91** | **6.49** | 3.01 | **3.47** | **5.27** |
| GPTQ | 4.00 | 5.49 | 7.20 | 4.00 | 4.78 | 6.34 | 4.00 | 3.35 | 5.15 |
| QuIP | – | – | – | 4.00 | 4.76 | 6.29 | 4.00 | 3.58 | 5.38 |
| SqueezeLLM | 4.05 | 5.23 | 6.78 | 4.04 | 4.67 | 6.15 | 4.03 | **3.20** | 5.04 |
| GPTVQ 1D | 4.06 | 5.27 | 6.83 | 4.06 | 4.67 | 6.17 | 4.03 | **3.20** | 5.04 |
| LNQ (Ours) | 4.05 | 5.26 | 6.82 | 4.04 | 4.67 | 6.17 | 4.03 | **3.20** | 5.04 |
| LNQ + GuidedQuant (Ours) | 4.05 | **5.21** | **6.75** | 4.04 | **4.65** | **6.14** | 4.03 | **3.20** | **5.03** |

(Gholami et al., 2024). Among weight-only techniques, three quantization formats are commonly used: *uniform scalar*, *non-uniform scalar*, and *vector* quantization (Frantar et al., 2023; Kim et al., 2024; Tseng et al., 2024b). With fixed bit-width constraints, non-uniform scalar quantization generally outperforms uniform scalar quantization, as its search space encompasses that of uniform scalar quantization. Meanwhile, vector quantization can outperform non-uniform scalar quantization by exploiting additional redundancies across weight dimensions.

Despite this, non-uniform scalar quantization offers advantages in inference latency. Table 2 compares end-to-end single-batch inference latency across these formats using the state-of-the-art GPU kernels: LUT-GEMM (Park et al., 2024a) for uniform scalar, Any-Precision-LLM (Park et al., 2024b) for non-uniform scalar, and QTIP (Tseng et al., 2024b) for vector quantization. Results show that vector quantization incurs higher latency due to its decoding overhead (Tseng et al., 2024b), whereas uniform and non-uniform scalar quantization have similar latency with minimal decoding overhead. Consequently, non-uniform scalar and vector quantization remain the primary formats of interest for weight-only quantization. In this context, we apply our GuidedQuant to both formats, achieving state-of-the-art performance in each.

For our experiments, we demonstrate the effectiveness of our method on the Llama-2 model family (Touvron et al., 2023), evaluating on 7B, 13B and 70B model. We use the RedPajama dataset (Computer, 2023) for calibration, following prior work (Egiazarian et al., 2024; Tseng et al., 2024a;b), with 1024 sentences, each containing 4096 tokens. We report perplexity on the WikiText2 (Merity et al., 2016) and C4 (Raffel et al., 2020) validation sets.

**Scalar Post-training Quantization Results.** We summarize the results of weight-only scalar quantization in Table 3, comparing our approach with GPTQ (Frantar et al., 2023), SqueezeLLM without mixed precision (Kim et al., 2024), QuIP (Chee et al., 2024), and GPTVQ 1D (van Baalen et al., 2024). For GPTQ and QuIP, we report the results from Egiazarian et al. (2024), which used the same or a larger calibration dataset, while for GPTVQ 1D, we reproduce the results with the same calibration data while adjusting the group size to align with the average bit-width for a fair comparison (see Appendix C.4 for details).

We evaluate the performance of LNQ both with and without the GuidedQuant objective. Notably, LNQ combined with GuidedQuant consistently outperforms all baselines across various bit-widths and model sizes. Additionally, LNQ with the layer-wise reconstruction objective surpasses GPTVQ 1D in all settings, demonstrating that our approach improves upon GPTVQ 1D by addressing its suboptimal optimization.

**Vector Post-training Quantization Results.** For vector post-training quantization (PTQ), we present the results in Table 4. We apply GuidedQuant to the state-of-the-art vector PTQ baseline, QTIP (Tseng et al., 2024b). We implement it on both the 1MAD and 3INST variants and report the variant that performs better among these two. Refer to

*Table 4.* Weight-only vector post-training quantization results *without fine-tuning* to the end-to-end loss. Wiki2 and C4 denotes perplexity on WikiText2 and C4, respectively. The perplexity is measured with the context size of 4096.

| | Llama-2-7B | | | Llama-2-13B | | | Llama-2-70B | | |
| Method | Bits↓ | Wiki2↓ | C4↓ | Bits↓ | Wiki2↓ | C4↓ | Bits↓ | Wiki2↓ | C4↓ |
|---|---|---|---|---|---|---|---|---|---|
| Original | 16 | 5.12 | 6.63 | 16 | 4.57 | 6.05 | 16 | 3.12 | 4.97 |
| GPTVQ 2D | 2.13 | 10.66 | 12.81 | 2.13 | 7.55 | 9.82 | 2.13 | 5.06 | 7.09 |
| GPTVQ 4D | 2.25 | 7.89 | 10.25 | 2.25 | 6.36 | 8.43 | 2.25 | 4.44 | 6.28 |
| QuIP# | 2.00 | 8.22 | 11.01 | 2.00 | 6.06 | 8.07 | 2.00 | 4.16 | 6.01 |
| AQLM | 2.02 | 6.59 | 8.54 | 2.19 | 5.37 | 7.16 | 2.07 | 3.94 | 5.72 |
| QTIP | 2.00 | 6.82 | 8.96 | 2.00 | 5.52 | 7.39 | 2.00 | 3.87 | 5.69 |
| QTIP + GuidedQuant (Ours) | 2.00 | **6.11** | **7.99** | 2.00 | **5.33** | **7.05** | 2.00 | **3.80** | **5.61** |
| GPTVQ 2D | 3.13 | 5.63 | 7.32 | 3.13 | 4.87 | 6.45 | 3.13 | 3.38 | 5.18 |
| QuIP# | 3.00 | 5.60 | 7.34 | 3.00 | 4.90 | 6.50 | 3.00 | 3.41 | 5.20 |
| AQLM | 3.04 | 5.46 | 7.08 | 3.03 | 4.82 | 6.37 | 3.01 | 3.36 | 5.17 |
| QTIP | 3.00 | 5.38 | 6.99 | 3.00 | 4.74 | 6.28 | 3.00 | 3.27 | 5.09 |
| QTIP + GuidedQuant (Ours) | 3.00 | **5.28** | **6.87** | 3.00 | **4.71** | **6.22** | 3.00 | **3.25** | **5.08** |
| GPTVQ 2D | 4.13 | 5.24 | 6.77 | 4.13 | 4.65 | 6.13 | 4.13 | 3.18 | 5.01 |
| QuIP# | 4.00 | 5.22 | 6.79 | 4.00 | 4.65 | 6.15 | 4.00 | 3.18 | 5.02 |
| AQLM | 4.04 | 5.21 | 6.75 | 3.94 | 4.65 | 6.14 | 4.14 | 3.19 | 5.03 |
| QTIP | 4.00 | 5.17 | 6.71 | 4.00 | 4.62 | 6.10 | 4.00 | 3.16 | **5.00** |
| QTIP + GuidedQuant (Ours) | 4.00 | **5.16** | **6.68** | 4.00 | **4.61** | **6.09** | 4.00 | **3.15** | **5.00** |

Appendix E.10 for results on different variants. We compare our approach with the following baselines: GPTVQ (van Baalen et al., 2024), QuIP# (Tseng et al., 2024a), AQLM (Egiazarian et al., 2024), and QTIP (Tseng et al., 2024b). For QuIP#, AQLM, and QTIP, we report the results from their respective papers, as they used the same or larger calibration datasets than ours. For GPTVQ, we report the reproduced results using our calibration data. Our method consistently outperforms all vector quantization baselines across different bit-widths and model sizes as well.

### 5.2. Weight-and-activation Quantization

Weight-and-activation quantization methods apply uniform quantization on both weights and activations to leverage the faster matrix multiplication units in the hardware (Ashkboos et al., 2024; Liu et al., 2024). State-of-the-art methods for weight-and-activation quantization include QuaRot (Ashkboos et al., 2024) and SpinQuant (Liu et al., 2024), which use rotation matrices to reduce the activation outliers before applying the uniform quantization. We incorporate our GuidedQuant objective into the weight quantization process of these methods, guiding the model to quantize the weights more accurately. Specifically, we implement GuidedQuant on top of the SpinQuant using GPTQ weight quantizer and present the results in Table 5. Following prior work, we use the WikiText2 dataset (Merity et al., 2016) for calibration, with 128 sentences, each containing 2048 tokens (Ashkboos et al., 2024; Liu et al., 2024). Our objective consistently improves the perplexity compared to the baseline methods,

*Table 5.* Weight-and-activation quantization results on Llama-2 models. L-2-7B, L-2-13B and L-2-70B denote Llama-2-7B, Llama-2-13B, and Llama-2-70B model, respectively. Wiki2 denotes perplexity on Wikitext2 with the context size of 2048.

| | | L-2-7B | L-2-13B | L-2-70B |
| Bits | Method | Wiki2↓ | Wiki2↓ | Wiki2↓ |
|---|---|---|---|---|
| 16 | Original | 5.47 | 4.88 | 3.32 |
| | QuaRot | 6.08 | 5.39 | 3.80 |
| W4A4KV4 | SpinQuant | 5.95 | 5.24 | **3.71** |
| | SpinQuant + GQuant (Ours) | **5.89** | **5.19** | **3.71** |
| | QuaRot | 6.02 | 5.34 | 3.77 |
| W4A4KV16 | SpinQuant | 5.90 | 5.22 | **3.68** |
| | SpinQuant + GQuant (Ours) | **5.84** | **5.17** | **3.68** |

demonstrating its effectiveness.

## 6. Conclusion

We introduced GuidedQuant, a novel PTQ approach that integrates gradient information from the end loss while preserving cross-weight dependencies within output channels. GuidedQuant improves state-of-the-art methods across quantization formats, including weight-only scalar, weight-only vector, and weight-and-activation quantization. Furthermore, we identified inefficiencies in the current state-of-the-art methods for non-uniform scalar quantization and proposed LNQ, a new algorithm that, when combined with GuidedQuant, improves over the state-of-the-art performance. These contributions advance the efficiency and accuracy of quantization for modern LLMs.

## Impact Statement

This work advances the compression of LLMs, in particular via post-training quantization. As discussed, quantization, and model compression more broadly, reduces the memory and computational requirements of LLMs and speeds up inference, thus reducing their environmental impact and enabling their use on resource-constrained devices and for latency-critical applications. This can also help democratize access to these models for organizations with limited resources and support privacy-preserving, offline applications. On the other hand, compression methods, including quantization, can adversely affect fairness in language models (Ramesh et al., 2023). While there are ongoing efforts to address fairness concerns in pruned LLMs (Zayed et al., 2024), extending these mitigation strategies to quantized models remains an important direction for future research. Furthermore, reducing the cost of using LLMs can also lower the barrier to their use by malicious actors. Finally, the energy and resources saved through compression might be reinvested elsewhere, so the net reduction in environmental harm is not guaranteed (Jevons paradox (Alcott, 2005)).

## Acknowledgements

This work was supported by Samsung Electronics Co., Ltd. (IO250418-12669-01), Mobile eXperience (MX) Business, Samsung Electronics Co., Ltd., Institute of Information & Communications Technology Planning & Evaluation (IITP) grant funded by the Korea government (MSIT) [No. RS-2020-II200882, (SW STAR LAB) Development of deployable learning intelligence via self-sustainable and trustworthy machine learning, No. RS-2021-II211343, Artificial Intelligence Graduate School Program (Seoul National University), and No. 2022-0-00480, RS-2022-II220480, Development of Training and Inference Methods for Goal-Oriented Artificial Intelligence Agents], and Basic Science Research Program through the National Research Foundation of Korea (NRF) funded by the Ministry of Education (RS-2023-00274280). Hyun Oh Song is the corresponding author.

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

## A. Additional Related Work

There's a large body of work on neural network compression, even when considering only quantization for LLMs, making a complete overview infeasible. Instead, we focus here on the works most related to ours.

**Hessian-based Compression**    Neural networks compression based on the second-order Taylor approximation of the end loss (Eq (2)) dates back to the early works of LeCun et al. (1989) and Hassibi & Stork (1992). OBD (LeCun et al., 1989) introduced this approach for pruning, under the assumption that the Hessian matrix is diagonal. OBS (Hassibi & Stork, 1992) improved upon this by dropping the diagonal assumption and instead approximating the Hessian by the empirical Fisher information matrix. However, applying OBS to large neural networks remains computationally intractable. To address this, various more efficient Hessian approximations have been proposed, including the K-FAC approximation (Martens & Grosse, 2015; Zeng & Urtasun, 2018; Wang et al., 2019; Tycho F. A. van der Ouderaa, 2024), block-diagonal Fisher approximation (Singh & Alistarh, 2020; Kurtic et al., 2022; Li et al., 2021), and diagonal Fisher approximation (Choi et al., 2017; Theis et al., 2018; Kim et al., 2024; Bai et al., 2024). Other strategies directly estimate inverse-Hessian vector products (Frantar et al., 2021). The most similar approaches to GuidedQuant are ones that employ block-diagonal Fisher approximation, which achieve a good trade-off between approximation accuracy and computation and storage cost. However, these methods remain intractable at the scale of modern LLMs (see Appendix E.11).

**Gradient-based Compression**    Various compression methods are based on a first-order Taylor approximation of the end loss, with respect to output feature maps or gates applied to them (Molchanov et al., 2017; 2019; You et al., 2019), or weights (Ding et al., 2019). The one most similar to GuidedQuant is (Molchanov et al., 2019), which employs the same criterion in Eq. (4) to prune filters and neurons in vision models. However, as explained in Section 3.2, adopting this criterion for quantizing modern LLMs is infeasible, without the averaging approximation we propose.

**Non-uniform Scalar PTQ for LLMs**    PTQ encompasses a vast array of work, so we focus on non-uniform scalar PTQ methods for LLMs that use look-up tables (codebooks) for weight decoding, which are closely related to our LNQ algorithm. One approach is zero-shot quantization, which requires no calibration data: Dynamic Tree Quantization (Dettmers et al., 2021) defines a new data type with dynamic exponential bits and stores decoded values in the codebook; Quantile Quantization (Dettmers & Zettlemoyer, 2023) saves quantile values of the weight distribution; and QLoRA (Dettmers et al., 2023) introduces the NF4 data type using quantiles of a standard normal distribution. These methods share a global codebook, with each layer maintaining its own scale parameters. HIGGS (Malinovskii et al., 2024b) further refines this by adopting MSE-optimal grids for the standard normal distribution and applying rotation matrices to approximate Gaussian weight distributions. Another line of work involves one-shot quantization methods that optimize the output quantization error using calibration data. For instance, SqueezeLLM (Kim et al., 2024) optimizes separate channel-wise codebooks via the $k$-means algorithm, while a 1D variant of GPTVQ (van Baalen et al., 2024) alternates between optimizing assignments with the GPTQ algorithm and refining codebooks with gradient descent. The GPTVQ 1D shows the strongest performance among this line of research. Although not a scalar PTQ method, the vector quantization variant of AQLM (Egiazarian et al., 2024) also follows a similar paradigm, optimizing assignments through CD and codebooks via gradient descent.

## B. Proofs

Here, we prove Remark 3.1 and Proposition 4.1, each restated here for convenience.

*Remark* 3.1. The sum of the layer-wise objective in Eq. (4) over all layers is equal to the following quadratic approximation of the change in the end loss

$$n \sum_{l=1}^{L} \sum_{j=1}^{d_{\text{out}}^{(l)}} (\mathbf{w}_j^{(l)} - \widehat{\mathbf{w}}_j^{(l)})^\top \mathbf{F}_j^{(l)} (\mathbf{w}_j^{(l)} - \widehat{\mathbf{w}}_j^{(l)}). \tag{5}$$

*Proof.*   Recall that $\mathbf{Z}^{(l)} = \mathbf{X}^{(l)} \mathbf{W}^{(l)}$. Then, by chain rule we have that $\frac{\partial \ell_i}{\partial \mathbf{w}_j^{(l)}} = \frac{\partial \ell_i}{\partial Z_{ij}^{(l)}} (\mathbf{X}_{i,:}^{(l)})^\top$. Note also that $\frac{\partial \ell}{\partial Z_{ij}^{(l)}} = \frac{\partial \ell_i}{\partial Z_{ij}^{(l)}}$.

Hence,

$$
\begin{aligned}
\left\| \frac{\partial \ell}{\partial \mathbf{Z}^{(l)}} \odot (\mathbf{X}^{(l)}\mathbf{W}^{(l)} - \mathbf{X}^{(l)}\widehat{\mathbf{W}}^{(l)}) \right\|_F^2 &= \sum_{j=1}^{d_{\text{out}}^{(l)}} \sum_{i=1}^{n} \left( \frac{\partial \ell_i}{\partial Z_{ij}^{(l)}} \mathbf{X}_{i,:}^{(l)}(\mathbf{w}_j^{(l)} - \widehat{\mathbf{w}}_j^{(l)}) \right)^2 \\
&= \sum_{j=1}^{d_{\text{out}}^{(l)}} \sum_{i=1}^{n} \left( \left( \frac{\partial \ell_i}{\partial \mathbf{w}_j^{(l)}} \right)^{\top} (\mathbf{w}_j^{(l)} - \widehat{\mathbf{w}}_j^{(l)}) \right)^2 \\
&= n \sum_{j=1}^{d_{\text{out}}^{(l)}} (\mathbf{w}_j^{(l)} - \widehat{\mathbf{w}}_j^{(l)})^{\top} \mathbf{F}_j^{(l)} (\mathbf{w}_j^{(l)} - \widehat{\mathbf{w}}_j^{(l)}),
\end{aligned}
$$

where the last equality follows from the definition of the Fisher blocks $\mathbf{F}_j^{(l)} = \frac{1}{n} \sum_{i=1}^{n} (\frac{\partial \ell_i}{\partial \mathbf{w}_j^{(l)}})(\frac{\partial \ell_i}{\partial \mathbf{w}_j^{(l)}})^{\top}$. Taking the sum over $l \in [L]$ on both sides yields the claim. $\qquad \square$

**Proposition 4.1.** *For any $j \in [d_{\text{out}}]$, let $f_j(\mathbf{c}, \mathbf{P}) = \|\mathbf{X}\mathbf{w}_j - \mathbf{X}\mathbf{P}\mathbf{c}\|_2^2$, and let $\mathbf{c}_t^{(j)}$ and $\mathbf{P}_t^{(j)}$ denote $\mathbf{c}^{(j)}$ and $\mathbf{P}^{(j)}$ at the $t$-th iteration of LNQ. Then, $f_j(\mathbf{c}_t^{(j)}, \mathbf{P}_t^{(j)}) \geq f_j(\mathbf{c}_{t+1}^{(j)}, \mathbf{P}_t^{(j)}) \geq f_j(\mathbf{c}_{t+1}^{(j)}, \mathbf{P}_{t+1}^{(j)})$ for all $t$, and the sequence $\{f_j(\mathbf{c}_t^{(j)}, \mathbf{P}_t^{(j)})\}_{t \geq 1}$ converges.*

*Proof.* We first show that the objective value is non-increasing in LNQ. For all $t \geq 1$, we have $\mathbf{P}_t^{(j)} \mathbf{1}_m = \mathbf{1}_{d_{\text{in}}}$ and thus the corresponding quantized weights $\widehat{\mathbf{w}}_j = \mathbf{P}_t^{(j)} \mathbf{c}_t^{(j)}$ are feasible Hence, CD is initialized with a feasible solution at each iteration $t$, so it acts as a descent method. Then,

$$
\begin{aligned}
f_j(\mathbf{c}_t^{(j)}, \mathbf{P}_t^{(j)}) &\geq f_j(\mathbf{c}_{t+1}^{(j)}, \mathbf{P}_t^{(j)}) && \text{(since } \mathbf{c}_{t+1}^{(j)} = \operatorname*{argmin}_{\mathbf{c}^{(j)} \in \mathbb{R}^m} f_j(\mathbf{c}, \mathbf{P}_t)) \\
&\geq f_j(\mathbf{c}_{t+1}^{(j)}, \mathbf{P}_{t+1}^{(j)}), && \text{(since CD does not increase the objective value)}
\end{aligned}
$$

for all $t \geq 1$. Since $f_j(\mathbf{c}, \mathbf{P})$ is bounded below by 0, the sequence $\{f_j(\mathbf{c}_t^{(j)}, \mathbf{P}_t^{(j)})\}$ is monotonically non-increasing and bounded below. Hence, it converges to its infimum by the monotone convergence theorem. $\qquad \square$

## C. Hyperparameters and Details

In this section, we clarify the hyperparameters and details of the methods discussed in the main paper.

### C.1. GuidedQuant

The proposed GuidedQuant method has a single hyperparameter: the number of group $g$ used to average the Hessian matrices $\overline{\mathbf{H}}_j$ (see Section 3.2). For weight-only quantization experiments, we set $g = 4$ for Llama-2-7B and Llama-2-13B, and $g = 2$ for Llama-2-70B. For weight-and-activation quantization experiments, we set $g = 1$. For the hyperparameter $g$, we selected the number of groups to be as large as possible within the limits of our computational and memory constraints. Notably, GuidedQuant also maintains strong performance with smaller values of $g$ (see Appendix E.5).

Computing the Hessian (Line 4 in Algorithm 1) and running the quantization algorithm $\mathcal{Q}$ (Line 5 in Algorithm 1) for each group and layer can be parallelized. We parallelize Hessian computation across groups. For quantization, we parallelize across groups in LNQ + GuidedQuant, while in QTIP + GuidedQuant and SpinQuant + GuidedQuant, we run this step in a sequential manner to minimally change the codebase of the original methods.

### C.2. LNQ

The proposed LNQ method has two hyperparameters: (1) the number of iterations during which we alternate between optimizing $\mathbf{c}$ and $\mathbf{P}$ ($T$ in Algorithm 2), and (2) the number of coordinate descent iterations over the output dimensions ($K$ in Algorithm 2). For Llama-2-7B and Llama-2-13B, we use $T = 2$ and $K = 4$, and for Llama-2-70B, we use $T = 1$ and $K = 4$ in all the experiments.

---

**Algorithm 3** Efficient CD algorithm with precomputation

---

**input** Hessian of the objective $\mathbf{H} \in \mathbb{R}^{d_{\text{in}} \times d_{\text{in}}}$, input weight $\mathbf{W} \in \mathbb{R}^{d_{\text{in}} \times d_{\text{out}}}$, current codebook $\mathbf{c}^{(j)} \in \mathbb{R}^m$ and current quantized weight $\widehat{\mathbf{W}} \in \mathbb{R}^{d_{\text{in}} \times d_{\text{out}}}$. Initialize $\mathbf{Q} \in \{1, \ldots, m\}^{d_{\text{in}} \times d_{\text{out}}}$ (rounded indices).

1: $\widetilde{\mathbf{H}} \leftarrow \text{diag}(\mathbf{H})^{-1}\mathbf{H}$, $\mathbf{U} \leftarrow \text{StrictUpper}(\widetilde{\mathbf{H}})$.

2: **for** $k = 1$ **to** $K$ **do**
3:     $\mathbf{B} \leftarrow \mathbf{U}(\widehat{\mathbf{W}} - \mathbf{W})$
4:     **for** $i = 1$ **to** $d_{\text{in}}$ **do**
5:        $\widehat{\mathbf{W}}_{i,:} \leftarrow \text{Round}(\mathbf{W}_{i,:} - \mathbf{B}_{i,:}),\ \mathbf{Q}_{i,:} \leftarrow \text{RoundIdx}(\mathbf{W}_{i,:} - \mathbf{B}_{i,:})$
6:        $\mathbf{B}_{(i+1):,:} \leftarrow \mathbf{B}_{(i+1):,:} + \mathbf{U}_{(i+1):,i}(\widehat{\mathbf{W}}_{i,:} - \mathbf{W}_{i,:})$
7:     **end for**
8: **end for**

9: $\forall i \in [d_{\text{in}}], j \in [d_{\text{out}}], q \in [m]: P_{iq}^{(j)} = \begin{cases} 1 & \text{if } q = Q_{ij}, \\ 0 & \text{otherwise.} \end{cases}$              {Extracting assignment matrix}

**output** $\mathbf{P}^{(1)}, \ldots, \mathbf{P}^{(d_{\text{out}})}$.

---

We further explain a derivation of the time complexity of the proposed LNQ algorithm (Algorithm 2), discussed in Section 4.2. First, the time complexity of the Cholesky decomposition for a matrix $\mathbf{H} \in \mathbb{R}^{d_{\text{in}} \times d_{\text{in}}}$ is $O(d_{\text{in}}^3)$ (Line 1).

For optimizing the codebook (Line 4 and 14), we analyze the computational cost within the loop as follows:

- Computing $\mathbf{L}^\top \mathbf{P}^{(j)}$ requires $O(d_{\text{in}}^2 m)$ time.

- Computing $\mathbf{L}^\top \mathbf{w}_j$ requires $O(d_{\text{in}}^2)$ time.

- `torch.linalg.lstsq` function uses QR decomposition of $\mathbf{L}^\top \mathbf{P}^{(j)}$ to compute least squares solution, which requires $O(d_{\text{in}} m^2)$ time.

Since $d_{\text{in}} \gg m$, the dominant cost is $O(d_{\text{in}}^2 m)$.

For computing $\hat{\mathbf{w}}_j = \mathbf{P}^{(j)} \mathbf{c}^{(j)}$ (Line 5), the cost is $O(d_{\text{in}} m)$.

In CD, the cost of the minimizing the objective for each coordinate $i$ (Line 8, Eq. (11)) is $O(d_{\text{in}} + m)$. Since $d_{\text{in}} \gg m$, the dominant cost is $O(d_{\text{in}})$. Considering loop iterations, optimizing the code (Lines 6-12) takes $O(d_{\text{in}}^2 K)$ time complexity.

Therefore, the cost of Lines 4-12 is $O(d_{\text{in}}^2(m + K))$, and the cost of Lines 2-15 is $O(d_{\text{in}}^2 d_{\text{out}} T(m + K))$. Including the Cholesky decomposition, the total time complexity of LNQ algorithm is $O(d_{\text{in}}^3 + d_{\text{in}}^2 d_{\text{out}} T(m + K))$.

### C.3. Efficient Implementation of CD Algorithm in LNQ

In the LNQ algorithm (Algorithm 2), computing the solution across all output channels $j \in [d_{\text{out}}]$ is independent and thus fully parallelizable. Therefore, we perform the coordinate descent (CD) updates for each output channel in parallel.

**Coordinate-wise Closed-form Solution.** For a given quantized weight matrix $\widehat{\mathbf{W}} \in \mathbb{R}^{d_{\text{in}} \times d_{\text{out}}}$, the CD update for the $i$-th input coordinate can be computed in parallel using the coordinate-wise closed-form solution as follows (Behdin et al., 2023, Lemma 1):

$$\widehat{\mathbf{W}}_{i,:} \leftarrow \text{Round}\left(\mathbf{W}_{i,:} - \frac{\mathbf{H}_{i,[d_{\text{in}}]\setminus i}}{H_{i,i}}\left(\widehat{\mathbf{W}}_{[d_{\text{in}}]\setminus i,:} - \mathbf{W}_{[d_{\text{in}}]\setminus i,:}\right)\right), \tag{12}$$

where $\text{Round}(\cdot) : \mathbb{R}^{1 \times d_{\text{out}}} \to \mathbb{R}^{1 \times d_{\text{out}}}$ rounds $j$-th element to the nearest point in the grid $\{c_1^{(j)}, \ldots, c_m^{(j)}\}$. We adopt this coordinate-wise closed-form solution within the CD loop.

**Precomputation Trick.** On GPUs, the coordinate-wise CD update in Eq. (12) can be accelerated by precomputing parts of the update that remain unchanged during previous coordinate updates. Specifically, when updating the $i$-th coordinate, the components of $\widehat{\mathbf{W}}$ corresponding to coordinates $(i + 1)$ to $d_{\text{in}}$ remain fixed and can therefore be precomputed before

**Algorithm 4** Efficient CD algorithm with precomputation and lazy batch-updates

---

**input** Hessian of the objective $\mathbf{H} \in \mathbb{R}^{d_{\text{in}} \times d_{\text{in}}}$, input weight $\mathbf{W} \in \mathbb{R}^{d_{\text{in}} \times d_{\text{out}}}$, current codebook $\mathbf{c}^{(j)} \in \mathbb{R}^m$ and current quantized weight $\widehat{\mathbf{W}} \in \mathbb{R}^{d_{\text{in}} \times d_{\text{out}}}$. Initialize $\mathbf{Q} \in \{1, \ldots, m\}^{d_{\text{in}} \times d_{\text{out}}}$ (rounded indices).

1: $\widetilde{\mathbf{H}} \leftarrow \text{diag}(\mathbf{H})^{-1}\mathbf{H}$ , $\mathbf{U} \leftarrow \text{StrictUpper}(\widetilde{\mathbf{H}})$.

2: **for** $k = 1$ **to** $K$ **do**

3: $\quad$ $\mathbf{B} \leftarrow \mathbf{U}(\widehat{\mathbf{W}} - \mathbf{W})$

4: $\quad$ **for** $s = 1,\ b+1,\ 2b+1, \ldots,\ d_{\text{in}} - b + 1$ **do**

5: $\quad\quad$ **for** $i = s$ **to** $s + b - 1$ **do**

6: $\quad\quad\quad$ $\widehat{\mathbf{W}}_{i,:} \leftarrow \text{Round}(\mathbf{W}_{i,:} - \mathbf{B}_{i,:}), \mathbf{Q}_{i,:} \leftarrow \text{RoundIdx}(\mathbf{W}_{i,:} - \mathbf{B}_{i,:})$

7: $\quad\quad\quad$ $\mathbf{B}_{(i+1):(s+b),:} \leftarrow \mathbf{B}_{(i+1):(s+b),:} + \mathbf{U}_{(i+1):(s+b),i}(\widehat{\mathbf{W}}_{i,:} - \mathbf{W}_{i,:})$

8: $\quad\quad$ **end for**

9: $\quad\quad$ $\mathbf{B}_{(s+b):,:} \leftarrow \mathbf{B}_{(s+b):,:} + \mathbf{U}_{(s+b):,s:(s+b)}(\widehat{\mathbf{W}}_{s:(s+b),:} - \mathbf{W}_{s:(s+b),:})$

10: $\quad$ **end for**

11: **end for**

12: $\forall i \in [d_{\text{in}}], j \in [d_{\text{out}}], q \in [m] : P_{iq}^{(j)} = \begin{cases} 1 & \text{if } q = Q_{ij}, \\ 0 & \text{otherwise.} \end{cases}$ $\qquad$ {Extracting assignment matrix}

**output** $\mathbf{P}^{(1)}, \ldots, \mathbf{P}^{(d_{\text{out}})}$.

---

entering the CD loop:

$$\mathbf{B} := \frac{\mathbf{H}_{i,[d_{\text{in}}]\setminus i}}{H_{i,i}} \left( \widehat{\mathbf{W}}_{[d_{\text{in}}]\setminus i,:} - \mathbf{W}_{[d_{\text{in}}]\setminus i,:} \right) = \underbrace{\frac{\mathbf{H}_{i,1:i}}{H_{i,i}} \left( \widehat{\mathbf{W}}_{1:i,:} - \mathbf{W}_{1:i,:} \right)}_{\text{Cannot be precomputed before the CD loop}} + \underbrace{\frac{\mathbf{H}_{i,(i+1):}}{H_{i,i}} \left( \widehat{\mathbf{W}}_{(i+1):,:} - \mathbf{W}_{(i+1):,:} \right)}_{\text{Can be precomputed before the CD loop}}.$$

To take advantage of this, we precompute the second term (which corresponds to future coordinates) for all $i \in [d_{\text{in}}]$ in parallel using matrix operations before entering the CD loop:

$$\mathbf{B} \leftarrow \text{StrictUpper}(\widetilde{\mathbf{H}})(\widehat{\mathbf{W}} - \mathbf{W}),$$

where $\widetilde{\mathbf{H}}$ is obtained by dividing each row of $\mathbf{H}$ by the corresponding diagonal entry $H_{i,i}$, and $\text{StrictUpper}(\cdot)$ extracts the strictly upper triangular part of the matrix.

During the CD loop, we use the precomputed $\mathbf{B}$ to compute the coordinate-wise update in Eq. (12), and after updating the $i$-th coordinate, we incrementally update $\mathbf{B}$ to reflect the new value of $\widehat{\mathbf{W}}_{i,:}$:

$$\widehat{\mathbf{W}}_{i,:} \leftarrow \text{Round}(\mathbf{W}_{i,:} - \mathbf{B}_{i,:})$$
$$\mathbf{B}_{(i+1):,:} \leftarrow \mathbf{B}_{(i+1):,:} + \text{StrictUpper}(\widetilde{\mathbf{H}})_{(i+1):,i}(\widehat{\mathbf{W}}_{i,:} - \mathbf{W}_{i,:}).$$

The full CD algorithm incorporating this precomputation strategy is provided in Algorithm 3.

This acceleration trick has been proposed in QuIP (Chee et al., 2024, Appendix B.2.) and QuantEase (Behdin et al., 2023). It is worth noting that this precomputation trick does not change the theoretical time complexity, but improves practical performance by exploiting the GPU parallelization. In particular, the CD update for the $i$-th coordinate in Equation (12) requires $2d_{\text{out}}(d_{\text{in}} - 1)$ FLOPs without precomputation, while with precomputation, the cost is reduced to $2d_{\text{out}}(d_{\text{in}} - i)$ FLOPs.

**Lazy Batch-updates.** After incorporating the precomputation trick, we observe that the update steps within the CD loop (Lines 4–7 in Algorithm 3) resemble the OBQ update scheme used in the GPTQ method (Frantar & Alistarh, 2022; Frantar et al., 2023). In OBQ, each iteration involves rounding a single coordinate and adjusts the not-yet-rounded coordinates accordingly. Analogously, our CD update with precomputation rounds $\mathbf{W}_{i,:} - \mathbf{B}_{i,:}$ for the $i$-th coordinate and incrementally updates $\mathbf{B}_{(i+1):,:}$ to reflect the new values of $\widehat{\mathbf{W}}_{i,:}$.

*Table 6.* Hyperparameters that we used in reproducing GPTVQ (van Baalen et al., 2024) in Table 3 and Table 4.

| Table | Weight bits | VQ dim | Codebook sharing group size | Scaling block size | Codebook bit-width | Avg bits |
|---|---|---|---|---|---|---|
| Table 3 | 2 | 1 | 1024 | – | 8 | 2.03 |
| | 3 | 1 | 2048 | – | 8 | 3.03 |
| | 4 | 1 | 8192 | 256 | 8 | 4.03 |
| | 4 | 1 | 4096 | 128 | 8 | 4.06 |
| Table 4 | 2 | 2 | 2048 | – | 8 | 2.13 |
| | 2 | 4 | 32768 | – | 8 | 2.25 |
| | 3 | 2 | 16384 | 64 | 8 | 3.13 |
| | 4 | 2 | 65536 | 64 | 8 | 4.13 |

Both OBQ and our CD update suffer from a low compute-to-memory ratio: although each iteration involves relatively few FLOPs, it requires frequent reading and writing to large matrices. As a result, these updates tend to be memory-bound and suffer from poor GPU utilization. To mitigate this, GPTQ introduces *lazy batch-updates*, in which a batch of coordinates (with batch size $b = 128$) is processed together. Within each batch, updates are applied sequentially to each coordinate, while corrections are made only for the remaining unprocessed coordinates within the batch. Once all $b$ coordinates in the batch are updated, a global correction step is performed for the rest of the matrix. This strategy improves memory efficiency by reducing the frequency of global updates.

We adopt this lazy batch-updates approach in our CD implementation with precomputation trick. Specifically, we restrict updates to the relevant portion of $\mathbf{B}$ within each block of $b$ coordinates, and defer global updates to $\mathbf{B}$ until the entire block has been processed. This significantly reduces memory-bound operations and enhances GPU utilization. The final efficient CD algorithm incorporating both precomputation trick and lazy batch-updates is given in Algorithm 4.

QuIP (Chee et al., 2024) also supports lazy batch-updates in their open-source code, though it is not mentioned in their paper. QuantEase (Behdin et al., 2023) does not use this approach in their implementation. As with the precomputation trick, lazy batch-updates do not change the theoretical time complexity. However, they substantially accelerate the overall algorithm in practice by better utilizing GPU resources.

**Speedup Factor**    To demonstrate the speedup achieved by our optimization techniques for the CD algorithm, we report the quantization time for quantizing the Llama-2-7B model into 4-bit precision on a single RTX 6000 Ada GPU. Without any optimizations, adopting the naive strategy of exhaustively evaluating the objective function for all coordinate choices and selecting the option with the lowest value takes 3.9 hours to quantize the entire model. Applying the coordinate-wise closed-form solution described in Eq. (12) reduces this time to 2.7 hours. Incorporating the precomputation trick further lowers it to 1.2 hours. Finally, applying lazy batch-updates brings the total quantization time down to just 0.9 hours. Overall, these optimizations yield more than a $4\times$ speedup in end-to-end quantization time on GPU.

### C.4. GPTVQ

In the original GPTVQ paper (van Baalen et al., 2024), the authors used 128 sentences from the WikiText2 dataset (Merity et al., 2016), each containing 2048 tokens, as a calibration data. For a fair comparison, we reproduced their method using their open-sourced code but used 1024 sentences of RedPajama dataset (Computer, 2023), each containing 4096 tokens. We adopted their default hyperparameters except for the group size and block size, which we adjusted to match the average bit width when comparing with different methods in Table 3. We provide a complete list of GPTVQ hyperparameters for each table in Table 6.

## D. Details on Experimental Setup

This section provides a detailed explanation of the experimental settings used.

*Table 7.* End-to-end inference throughput of Llama-2 models on RTX 4090 GPU, including the vector quantization kernel *after* fusing the query/key/value projection matrices into one linear layer and the up/gate projection matrices into another when measuring the throughput. OOM indicates an Out-of-Memory error, meaning the GPU lacks memory to run model inference.

| | Llama-2-7B | | Llama-2-13B | | Llama-2-70B | |
|---|---|---|---|---|---|---|
| Type | Bits↓ | Tok/s↑ | Bits↓ | Tok/s↑ | Bits↓ | Tok/s↑ |
| Original | 16 | 67 | 16 | OOM | 16 | OOM |
| Uniform scalar | 2.00 | 334 | 2.00 | 200 | 2.00 | 47 |
| Non-uniform scalar | 2.01 | 347 | 2.01 | 203 | 2.01 | 47 |
| Vector | 2.00 | 200 | 2.00 | 121 | 2.00 | 38 |
| Vector *(fused)* | 2.00 | 248 | 2.00 | 153 | 2.00 | 42 |
| Uniform scalar | 3.00 | 260 | 3.00 | 150 | 3.00 | OOM |
| Non-uniform scalar | 3.03 | 264 | 3.02 | 148 | 3.01 | OOM |
| Vector | 3.00 | 176 | 3.00 | 103 | 3.00 | OOM |
| Vector *(fused)* | 3.00 | 209 | 3.00 | 123 | 3.00 | OOM |
| Uniform scalar | 4.00 | 214 | 4.00 | 121 | 4.00 | OOM |
| Non-uniform scalar | 4.05 | 209 | 4.04 | 116 | 4.03 | OOM |
| Vector | 4.00 | 151 | 4.00 | 89 | 4.00 | OOM |
| Vector *(fused)* | 4.00 | 176 | 4.00 | 103 | 4.00 | OOM |

### D.1. End-to-end Inference Throughput Experiments (Table 2)

In Table 2, we measure each model's inference throughput in generating 100 tokens on RTX 4090 GPU, after integrating the kernels with into a PyTorch-based inference pipeline optimized with the `torch.compile` function (Ansel et al., 2024; Gray, 2019). For QTIP, our chosen vector quantization kernel, we adopt the HYB variant of it as its GPU kernel is publicly available, though it is possible to implement fast GPU kernels with other variants as well (Tseng et al., 2024b).

For the base model and for models quantized using uniform or non-uniform scalar formats, we fuse the query/key/value projection matrices into one linear layer and the up/gate projection matrices into another when measuring the throughput. This fusion trick can be applied to QTIP as well, provided the matrices are fused before quantization and the scale parameters are shared across layers. However, in the main paper, we present QTIP results without fusion to match the original experimental setup (and reported numbers) from their work, in which they quantize the layers independently without fusing them. Meanwhile, scalar quantization methods quantize the layer in an output channel-wise manner, and this allows fusing matrices even when layers are quantized separately.

For completeness, we include Table 7, which also shows QTIP's fused end-to-end throughput (measured using dummy values) to illustrate the impact of fusion, restating the relevant results from Table 2. Although the fusion boosts the throughput, it does not change the conclusion that QTIP still runs more slowly than the scalar quantization methods.

### D.2. Implementation Details for Different Quantization Types

GuidedQuant employs a quantization algorithm $\mathcal{Q}$ as a subroutine (Line 8 in Algorithm 1). In this section, we clarify which specific quantization algorithm $\mathcal{Q}$ each method uses, which GuidedQuant builds upon in Algorithm 1. We integrate GuidedQuant with three different quantization methods: (1) LNQ for weight-only scalar quantization, (2) QTIP for weight-only vector quantization, and (3) SpinQuant for weight-and-activation quantization. LNQ adopts the algorithm shown in Algorithm 2, QTIP uses the BlockLDLQ algorithm proposed in Tseng et al. (2024a), and SpinQuant employs the GPTQ algorithm introduced in Frantar et al. (2023).

*Table 8.* Total GPU cost incurred during the quantization process for LNQ and QTIP, both with and without GuidedQuant, across various group sizes $g$. We specify the number and type of GPU used in the parentheses. R6A denotes the RTX 6000 Ada GPU.

| | | LNQ | | | QTIP | | |
|---|---|---|---|---|---|---|---|
| Model | Method | GPU Cost - 2 bits | GPU Cost - 3 bits | GPU Cost - 4 bits | GPU Cost - 2 bits | GPU Cost - 3 bits | GPU Cost - 4 bits |
| Llama-2-7B | Layer-wise (LNQ, QTIP) | 0.5 h (1×R6A) | 0.6 h (1×R6A) | 0.9 h (1×R6A) | 1.3 h (1×R6A) | 1.2 h (1×R6A) | 1.2 h (1×R6A) |
| | Layer-wise + GQuant ($g = 1$) | 0.5 h (1×R6A) | 0.6 h (1×R6A) | 0.9 h (1×R6A) | 1.3 h (1×R6A) | 1.2 h (1×R6A) | 1.2 h (1×R6A) |
| | Layer-wise + GQuant ($g = 2$) | 0.6 h (1×R6A) | 0.7 h (1×R6A) | 0.9 h (1×R6A) | 1.5 h (1×R6A) | 1.4 h (1×R6A) | 1.5 h (1×R6A) |
| | Layer-wise + GQuant ($g = 4$) | 0.7 h (1×R6A) | 0.7 h (1×R6A) | 0.9 h (1×R6A) | 1.9 h (1×R6A) | 1.9 h (1×R6A) | 1.9 h (1×R6A) |
| Llama-2-13B | Layer-wise (LNQ, QTIP) | 0.9 h (1×R6A) | 1.1 h (1×R6A) | 1.6 h (1×R6A) | 3.0 h (1×R6A) | 2.7 h (1×R6A) | 2.7 h (1×R6A) |
| | Layer-wise + GQuant ($g = 1$) | 0.9 h (1×R6A) | 1.1 h (1×R6A) | 1.6 h (1×R6A) | 3.0 h (1×R6A) | 2.7 h (1×R6A) | 2.7 h (1×R6A) |
| | Layer-wise + GQuant ($g = 2$) | 1.1 h (1×R6A) | 1.2 h (1×R6A) | 1.6 h (1×R6A) | 2.4 h (1×R6A) | 2.2 h (1×R6A) | 2.3 h (1×R6A) |
| | Layer-wise + GQuant ($g = 4$) | 1.2 h (1×R6A) | 1.3 h (1×R6A) | 1.7 h (1×R6A) | 3.0 h (1×R6A) | 3.0 h (1×R6A) | 3.0 h (1×R6A) |
| Llama-2-70B | Layer-wise (LNQ, QTIP) | 2.6 h (1×R6A) | 3.3 h (1×R6A) | 5.1 h (1×R6A) | 12.0 h (1×R6A) | 10.8 h (1×R6A) | 11.0 h (1×R6A) |
| | Layer-wise + GQuant ($g = 1$) | 2.6 h (1×R6A) | 3.3 h (1×R6A) | 5.1 h (1×R6A) | 12.0 h (1×R6A) | 10.8 h (1×R6A) | 11.0 h (1×R6A) |
| | Layer-wise + GQuant ($g = 2$) | 3.7 h (1×R6A) | 4.7 h (1×R6A) | 6.8 h (1×R6A) | 13.0 h (1×R6A) | 11.9 h (1×R6A) | 12.0 h (1×R6A) |

*Table 9.* Total GPU cost and disk usage incurred during the gradient and Hessian caching processes for each objective—weighted $k$-means (SqueezeLLM), layer-wise (LNQ, QTIP), and GuidedQuant. We specify the number and type of GPU used in the parentheses. R6A and A100 denote the RTX 6000 Ada GPU and the A100 GPU, respectively. The calibration data are 1024 sentences of the RedPajama dataset, each containing 4096 tokens.

| | | Gradient Caching | | Hessian Caching | |
|---|---|---|---|---|---|
| Model | Method | GPU Cost | Disk Size | GPU Cost | Disk Size |
| Llama-2-7B | Weighted $k$-means (SqueezeLLM) | 0.3 h (1×A100) | 13 GiB | – | – |
| | Layer-wise (LNQ, QTIP) | – | – | 0.3 h (4×R6A) | 27 GiB |
| | Layer-wise + GQuant ($g = 1$) | 0.3 h (1×A100) | 2 GiB | 0.3 h (4×R6A) | 27 GiB |
| | Layer-wise + GQuant ($g = 2$) | 0.3 h (1×A100) | 4 GiB | 0.4 h (4×R6A) | 53 GiB |
| | Layer-wise + GQuant ($g = 4$) | 0.3 h (1×A100) | 7 GiB | 0.8 h (4×R6A) | 106 GiB |
| Llama-2-13B | Weighted $k$-means (SqueezeLLM) | 0.6 h (2×A100) | 25 GiB | – | – |
| | Layer-wise (LNQ, QTIP) | – | – | 0.5 h (4×R6A) | 52 GiB |
| | Layer-wise + GQuant ($g = 1$) | 0.6 h (2×A100) | 3 GiB | 0.5 h (4×R6A) | 52 GiB |
| | Layer-wise + GQuant ($g = 2$) | 0.6 h (2×A100) | 7 GiB | 0.9 h (4×R6A) | 104 GiB |
| | Layer-wise + GQuant ($g = 4$) | 0.6 h (2×A100) | 13 GiB | 1.5 h (4×R6A) | 208 GiB |
| Llama-2-70B | Weighted $k$-means (SqueezeLLM) | 2.7 h (6×A100) | 129 GiB | – | – |
| | Layer-wise (LNQ, QTIP) | – | – | 3.5 h (4×R6A) | 366 GiB |
| | Layer-wise + GQuant ($g = 1$) | 2.7 h (6×A100) | 5 GiB | 3.5 h (4×R6A) | 366 GiB |
| | Layer-wise + GQuant ($g = 2$) | 2.7 h (6×A100) | 9 GiB | 5.8 h (4×R6A) | 731 GiB |

# E. Additional Results and Discussions

### E.1. Quantization Cost

In this section, we present a detailed breakdown of the computational costs associated with our method, as summarized in Table 8 and Table 9. The layer-wise quantization methods on which we build typically require two phases: (1) caching the Hessian matrices to disk, and (2) loading them to quantize weights based on these cached Hessian matrices. It is worth noting that the cost of the first phase (caching) can be amortized if one needs to quantize the same model multiple times at different bit-widths or configurations, as the Hessian matrices can be reused. We report the weight quantization cost in Table 8, and the Hessian-caching cost in Table 9.

From Table 8, observe that for $g = 1$, the quantization is identical to standard layer-wise quantization, since the Hessian size is the same. Even for $g = 2$ or $g = 4$, the quantization cost does not increase by more than 50%. This is because while more Hessian matrices are employed, each weight block to be quantized becomes correspondingly smaller, leaving the total computation unchanged. All these steps can be performed in an embarrassingly parallel manner; for example, quantizing Llama-2-70B using our LNQ algorithm takes less than three hours when using 8 RTX 6000 Ada GPUs.

We further report the cost of caching Hessian matrices in Table 9, along with the number of GPU used and the disk size requirements. Note that while we used 4 GPUs for caching, this process is also fully parallelizable; using fewer GPUs will simply take longer (it can run on a single GPU), whereas additional GPUs can shorten the total time. Finally, our method's

*Table 10.* Weight-only scalar post-training quantization results on Llama-3 models. Wiki2 and C4 denotes perplexity on WikiText2 and C4, respectively. The perplexity is measured with the context size of 8192.

| | Llama-3-8B | | | Llama-3-70B | | |
|---|---|---|---|---|---|---|
| Method | Bits↓ | Wiki2↓ | C4↓ | Bits↓ | Wiki2↓ | C4↓ |
| Original | 16 | 5.54 | 7.10 | 16 | 2.59 | 5.78 |
| SqueezeLLM | 2.01 | 16322 | 1501 | 2.01 | 38.53 | 38.15 |
| LNQ (Ours) | 2.01 | 133.00 | 72.75 | 2.01 | 24.22 | 19.71 |
| LNQ + GuidedQuant (Ours) | 2.01 | **30.80** | **20.41** | 2.01 | **10.21** | **11.06** |
| SqueezeLLM | 3.03 | 7.39 | 8.84 | 3.02 | 4.12 | 6.44 |
| LNQ (Ours) | 3.03 | 7.28 | 8.46 | 3.01 | 4.57 | 6.61 |
| LNQ + GuidedQuant (Ours) | 3.03 | **6.99** | **8.10** | 3.01 | **3.90** | **6.27** |
| SqueezeLLM | 4.05 | 5.91 | 7.43 | 4.03 | 2.91 | 5.91 |
| LNQ (Ours) | 4.05 | 5.90 | 7.40 | 4.03 | 3.05 | 5.94 |
| LNQ + GuidedQuant (Ours) | 4.05 | **5.80** | **7.32** | 4.03 | **2.89** | **5.89** |

*Table 11.* Weight-only scalar post-training quantization results on Llama-2 models, including end-to-end throughput. Wiki2 and C4 denotes perplexity on WikiText2 and C4, respectively. The perplexity is measured with the context size of 4096. Throughput is evaluated on an RTX 3090 GPU, reported as the average of 5 runs with standard deviation in parentheses. OOM indicates an Out-of-Memory error, meaning the GPU lacks memory to run model inference.

| | Llama-2-7B | | | | Llama-2-13B | | | | Llama-2-70B | | | |
|---|---|---|---|---|---|---|---|---|---|---|---|---|
| Method | Bits↓ | Wiki2↓ | C4↓ | Tok/s↑ | Bits↓ | Wiki2↓ | C4↓ | Tok/s↑ | Bits↓ | Wiki2↓ | C4↓ | Tok/s↑ |
| Original | 16 | 5.12 | 6.63 | 64.8 (0.1) | 16 | 4.57 | 6.05 | OOM | 16 | 3.12 | 4.97 | OOM |
| SqueezeLLM | 2.01 | 39.58 | 44.05 | 245.1 (1.8) | 2.01 | 16.24 | 19.20 | 140.5 (0.5) | 2.01 | 9.17 | 13.03 | 31.5 (0.0) |
| LNQ (Ours) | 2.01 | 23.31 | 26.71 | 244.6 (0.6) | 2.01 | 8.78 | 11.80 | 141.1 (0.4) | 2.01 | 5.23 | 7.31 | 31.6 (0.1) |
| LNQ + GuidedQuant (Ours) | 2.01 | **8.83** | **11.15** | 244.4 (2.9) | 2.01 | **7.26** | **9.17** | 141.2 (0.5) | 2.01 | **5.04** | **7.04** | 31.6 (0.1) |
| SqueezeLLM | 3.03 | 5.74 | 7.44 | 207.3 (1.6) | 3.02 | 4.99 | 6.60 | 118.0 (0.5) | 3.01 | 3.53 | 5.31 | OOM |
| LNQ (Ours) | 3.03 | 5.89 | 7.74 | 207.3 (2.1) | 3.02 | 5.02 | 6.68 | 118.0 (0.6) | 3.01 | 3.50 | 5.31 | OOM |
| LNQ + GuidedQuant (Ours) | 3.03 | **5.57** | **7.22** | 207.6 (1.7) | 3.02 | **4.91** | **6.49** | 117.9 (0.6) | 3.01 | **3.47** | **5.27** | OOM |
| SqueezeLLM | 4.05 | 5.23 | 6.78 | 161.8 (1.5) | 4.04 | 4.67 | 6.15 | 89.8 (0.1) | 4.03 | **3.20** | 5.04 | OOM |
| LNQ (Ours) | 4.05 | 5.26 | 6.82 | 161.7 (1.6) | 4.04 | 4.67 | 6.17 | 89.7 (0.1) | 4.03 | **3.20** | 5.04 | OOM |
| LNQ + GuidedQuant (Ours) | 4.05 | **5.21** | **6.75** | 162.0 (1.8) | 4.04 | **4.65** | **6.14** | 89.8 (0.1) | 4.03 | **3.20** | **5.03** | OOM |

disk-space requirement is proportional to the number of groups $g$. However, we highlight that for constrained disk space, choosing a smaller number of groups can still capture most of the performance benefits (Table 13).

### E.2. Results on Llama-3 Models

In this section, we present the results of evaluating LNQ and LNQ combined with GuidedQuant on Llama-3-8B and Llama-3-70B models, comparing with SqueezeLLM under a weight-only scalar quantization setting. We present the results in Table 10. We use RedPajama dataset (Computer, 2023) for calibration with 1024 sentences, each containing 4096 tokens. We set the number of groups to be $g = 1$ for Llama-3-8B and Llama-3-70B, and set the hyperparameters for LNQ (and LNQ + GuidedQuant) to be $T = 2, K = 4$ for Llama-3-8B and $T = 1, K = 4$ for Llama-3-70B model. LNQ with GuidedQuant consistently outperforms the baselines, demonstrating the robustness and effectiveness of our approach.

### E.3. Additional Inference Throughput Results

GuidedQuant leverages existing CUDA kernels (Any-Precision-LLM kernel (Park et al., 2024b) for weight-only scalar and QTIP kernel (Tseng et al., 2024b) for weight-only vector quantization) and optimizes assignment and codebook values, thus achieving improved performance without sacrificing inference throughput. To validate this, we compare weight-only scalar PTQ results on Llama-2 models across methods using the same CUDA kernel, as shown in Table 11. Specifically, we report perplexity and end-to-end throughput for SqueezeLLM, LNQ, and LNQ + GuidedQuant, all using the Any-Precision Kernel

*Table 12.* Weight-only scalar post-training quantization results, evaluated on zero-shot and few-shot downstream tasks. Zero-shot Avg denotes the average accuracy across eight zero-shot tasks: BoolQ, PIQA, SIQA, HellaSwag, WinoGrande, ARC-easy, ARC-challenge, and OBQA. For the few-shot benchmark, MMLU (5-shot) denotes accuracy on the MMLU benchmark in a 5-shot setting. We report the standard error in parentheses and bold the best results, as well as those whose accuracy score falls within the top score ± standard error.

| | Llama-2-7B | | | Llama-2-13B | | |
|---|---|---|---|---|---|---|
| Method | Bits↓ | Zero-shot Avg↑ | MMLU (5-shot)↑ | Bits↓ | Zero-shot Avg↑ | MMLU (5-shot)↑ |
| Original | 16 | 59.88 (0.43) | 45.97 (0.41) | 16 | 62.80 (0.43) | 54.93 (0.40) |
| SqueezeLLM | 2.01 | 41.80 (0.41) | 24.75 (0.36) | 2.01 | 42.44 (0.41) | 24.47 (0.36) |
| GPTVQ 1D | 2.03 | 37.35 (0.40) | 26.56 (0.37) | 2.03 | 46.34 (0.41) | 29.63 (0.38) |
| LNQ (Ours) | 2.01 | 40.30 (0.40) | 26.76 (0.37) | 2.01 | 49.51 (0.42) | 32.51 (0.39) |
| LNQ + GuidedQuant (Ours) | 2.01 | **50.39 (0.43)** | **31.53 (0.39)** | 2.01 | **53.98 (0.43)** | **40.15 (0.41)** |
| SqueezeLLM | 3.03 | 57.55 (0.43) | 40.59 (0.41) | 3.02 | **61.16 (0.43)** | 49.94 (0.40) |
| GPTVQ 1D | 3.03 | 54.92 (0.43) | 41.08 (0.41) | 3.03 | 60.38 (0.43) | 52.06 (0.40) |
| LNQ (Ours) | 3.03 | 56.85 (0.43) | 42.18 (0.41) | 3.02 | 60.61 (0.43) | 51.62 (0.40) |
| LNQ + GuidedQuant (Ours) | 3.03 | **58.16 (0.43)** | **43.38 (0.41)** | 3.02 | **61.00 (0.43)** | **52.67 (0.40)** |
| SqueezeLLM | 4.05 | **59.41 (0.43)** | **44.79 (0.41)** | 4.04 | **62.32 (0.43)** | 54.52 (0.40) |
| GPTVQ 1D | 4.06 | **59.23 (0.43)** | **45.06 (0.41)** | 4.06 | **62.37 (0.43)** | **54.95 (0.40)** |
| LNQ (Ours) | 4.05 | **59.14 (0.43)** | 44.51 (0.41) | 4.04 | **62.40 (0.43)** | **54.79 (0.40)** |
| LNQ + GuidedQuant (Ours) | 4.05 | **59.41 (0.43)** | **45.16 (0.41)** | 4.04 | **62.17 (0.43)** | 54.39 (0.40) |

([Park et al.](), [2024b]()). Throughput is measured on an RTX 3090 GPU as the average of 5 runs, with standard deviation in parentheses. Results confirm that our methods (LNQ and LNQ + GuidedQuant) achieve better perplexity while maintaining the same throughput as other method using the identical kernel.

### E.4. Evaluations on Zero-shot and Few-shot Downstream Benchmarks

In this section, we provide the evaluations on zero-shot and few-shot downstream tasks of our methods (LNQ and LNQ + GuidedQuant) alongside baselines (SqueezeLLM and GPTVQ 1D) under the weight-only scalar quantization settings, using Llama-2-7B and Llama-2-13B models, in Table 12. The evaluation includes eight zero-shot tasks: BoolQ ([Clark et al.](), [2019]()), PIQA ([Bisk et al.](), [2020]()), SIQA ([Sap et al.](), [2019]()), HellaSwag ([Zellers et al.](), [2019]()), WinoGrande ([Sakaguchi et al.](), [2019]()), ARC-easy ([Clark et al.](), [2018]()), ARC-challenge ([Clark et al.](), [2018]()), and OBQA ([Mihaylov et al.](), [2018]()). For a few-shot benchmark, we include results on the MMLU ([Hendrycks et al.](), [2021]()) benchmark in a 5-shot setting. We evaluate on these tasks using version 0.4.3 of the `lm-evaluation-harness` library ([Gao et al.](), [2024]()).

Table 12 reports both accuracy and standard error for all methods. We highlight the best-performing results, as well as those whose accuracy falls within the top score ± standard error, under the same bit width constraint. The results show that LNQ combined with GuidedQuant consistently matches or surpasses baseline performance, with notable improvements in extreme quantization scenarios, such as 2-bit quantization.

### E.5. Results on Varying the Number of Groups $g$

In this section, we present results on how varying the number of groups $g$ (introduced in Section 3.2) affects performance, focusing on whether fewer groups preserve accuracy or introduce trade-offs when averaging the Hessian within each group. Table 13 summarizes the impact of changing $g$ under a non-uniform scalar quantization scheme. While increasing $g$ can moderately improve results in extreme cases (e.g., quantizing models into 2 bits), performance differences across the number of groups remain minimal in other scenarios. Note that for weight-only quantization experiments, we chose $g = 4$ for Llama-2-7B and Llama-2-13B, and $g = 2$ for Llama-2-70B. Still, smaller number of groups are sufficient for achieving most of the performance gains, making them a practical choice for resource-constrained scenarios.

### E.6. Ablation Study on Assignments Optimization in LNQ

In this section, we evaluate our choice of using cyclic CD algorithm instead of GPTQ to solve Problem (8) for a fixed codebook $\mathbf{c}^{(j)}$ in LNQ. In particular, we compare two variants of LNQ with the GuidedQuant objective: the variant described

*Table 13.* Results with different number of groups $g$ in weight-only post-training quantization results on non-uniform scalar quantization format, *without fine-tuning* to the end-to-end loss. Wiki2 and C4 denotes perplexity on WikiText2 and C4, respectively, which are measured with the context size of 4096.

| Method | Number of groups $g$ | Llama-2-7B | | | Llama-2-13B | | | Llama-2-70B | | |
|---|---|---|---|---|---|---|---|---|---|---|
| | | Bits↓ | Wiki2↓ | C4↓ | Bits↓ | Wiki2↓ | C4↓ | Bits↓ | Wiki2↓ | C4↓ |
| Original | – | 16 | 5.12 | 6.63 | 16 | 4.57 | 6.05 | 16 | 3.12 | 4.97 |
| LNQ | – | 2.01 | 23.31 | 26.71 | 2.01 | 8.78 | 11.80 | 2.01 | 5.23 | 7.31 |
| LNQ + GuidedQuant | 1 | 2.01 | 9.00 | 11.35 | 2.01 | 7.32 | 9.29 | 2.01 | 5.11 | 7.06 |
| | 2 | 2.01 | 8.82 | 11.20 | 2.01 | 7.18 | 9.22 | 2.01 | 5.04 | 7.04 |
| | 4 | 2.01 | 8.83 | 11.15 | 2.01 | 7.26 | 9.17 | – | – | – |
| LNQ | – | 3.03 | 5.89 | 7.74 | 3.02 | 5.02 | 6.68 | 3.01 | 3.50 | 5.31 |
| LNQ + GuidedQuant | 1 | 3.03 | 5.55 | 7.23 | 3.02 | 4.92 | 6.49 | 3.01 | 3.46 | 5.27 |
| | 2 | 3.03 | 5.57 | 7.22 | 3.02 | 4.92 | 6.49 | 3.01 | 3.47 | 5.27 |
| | 4 | 3.03 | 5.57 | 7.22 | 3.02 | 4.91 | 6.49 | – | – | – |
| LNQ | – | 4.05 | 5.26 | 6.82 | 4.04 | 4.67 | 6.17 | 4.03 | 3.20 | 5.04 |
| LNQ + GuidedQuant | 1 | 4.05 | 5.21 | 6.75 | 4.04 | 4.65 | 6.14 | 4.03 | 3.20 | 5.03 |
| | 2 | 4.05 | 5.22 | 6.75 | 4.04 | 4.65 | 6.14 | 4.03 | 3.20 | 5.03 |
| | 4 | 4.05 | 5.21 | 6.75 | 4.04 | 4.65 | 6.14 | – | – | – |

*Table 14.* Ablation study on optimizing discrete assignment $\mathbf{P}$ in Problem (8). We compare two algorithms for optimizing discrete assignments; GPTQ and coordinate descent algorithm. Wiki2 and C4 denotes perplexity on WikiText2 and C4, respectively, which are measured with the context size of 4096.

| Method | Optimization method for $\mathbf{P}$ | Llama-2-7B | | | Llama-2-13B | | | Llama-2-70B | | |
|---|---|---|---|---|---|---|---|---|---|---|
| | | Bits↓ | Wiki2↓ | C4↓ | Bits↓ | Wiki2↓ | C4↓ | Bits↓ | Wiki2↓ | C4↓ |
| Original | – | 16 | 5.12 | 6.63 | 16 | 4.57 | 6.05 | 16 | 3.12 | 4.97 |
| LNQ + GQuant | GPTQ | 2.01 | 9.65 | 11.83 | 2.01 | 7.96 | 11.65 | 2.01 | **4.92** | **6.93** |
| | Coordinate Descent | 2.01 | **8.83** | **11.15** | 2.01 | **7.26** | **9.17** | 2.01 | 5.04 | 7.04 |
| LNQ + GQuant | GPTQ | 3.03 | 5.58 | 7.25 | 3.02 | **4.91** | 6.50 | 3.01 | **3.47** | **5.27** |
| | Coordinate Descent | 3.03 | **5.57** | **7.22** | 3.02 | **4.91** | **6.49** | 3.01 | **3.47** | **5.27** |
| LNQ + GQuant | GPTQ | 4.05 | 5.22 | **6.75** | 4.04 | **4.65** | **6.14** | 4.03 | **3.20** | **5.03** |
| | Coordinate Descent | 4.05 | **5.21** | **6.75** | 4.04 | **4.65** | **6.14** | 4.03 | **3.20** | **5.03** |

in Section 4.2, which updates the assignments using cyclic CD, and alternative variant that uses GPTQ for assignments updates. Both variants update the codebook using the closed-form solution in (9). We report the results on Llama-2-7B model, evaluated on WikiText2 and C4 datasets, in Table 14. Our experiments show that CD consistently outperforms or matches GPTQ, validating our choice of using CD to optimize the assignment matrix $\mathbf{P}^{(j)}$.

### E.7. End-to-end Fine-tuning Results

Recent weight-only quantization methods have explored fine-tuning quantized models using extensive data and compute to improve performance for low-bit models (Tseng et al., 2024a;b; Malinovskii et al., 2024a). In Table 15, we summarize the performance of quantized models after further fine-tuning on end loss using more data and compute for scalar weight-only quantization. We implement PV-Tuning (Malinovskii et al., 2024a) in non-uniform scalar quantization setting and report the performance of both our model and SqueezeLLM after fine-tuning with it. For SqueezeLLM and LNQ + GuidedQuant, we obtain the results using the official open-source implementation of PV-Tuning. Our fine-tuning setup uses training data from RedPajama dataset (Computer, 2023), with a context size of 4096 tokens, a batch size of 128 sentences, and fine-tuning for 128 steps in 2-bit quantization and 32 steps in 3-bit quantization. For GPTQ (uniform scalar quantization), we report the results from the PV-Tuning paper (Malinovskii et al., 2024a).

*Table 15.* Weight-only quantization results on Llama-2-7B model *after fine-tuning* with end-to-end loss. For scalar quantization methods, we report the performance after fine-tuning with PV-Tuning (Malinovskii et al., 2024a).

| Type | Method | Bits↓ | Wiki2↓ | C4↓ |
|------|--------|-------|--------|-----|
| Type | Original | 16 | 5.12 | 6.63 |
| Weight-only Scalar | GPTQ | 2.14 | 8.43 | 10.82 |
| | SqueezeLLM | 2.01 | 6.78 | 8.82 |
| | LNQ + GQuant (Ours) | 2.01 | **6.53** | **8.53** |
| | SqueezeLLM | 3.03 | 5.53 | 7.23 |
| | LNQ + GQuant (Ours) | 3.03 | **5.50** | **7.14** |

*Table 16.* Weight-and-activation quantization results on Llama-2-7B model, while quantizing weights into 2- and 3-bits. Wiki2 denotes perplexity on Wikitext2 with the context size of 2048. W$x$A$y$KV$z$ indicates quantizing weights into $x$-, activations into $y$-, and KV cache to $z$-bits, respectively.

| Method | Bits↓ | Wiki2↓ |
|--------|-------|--------|
| Original | 16 | 5.12 |
| SpinQuant | W2A4KV4 | 100.22 |
| SpinQuant + GQuant (Ours) | W2A4KV4 | **36.05** |
| SpinQuant | W3A4KV4 | 6.61 |
| SpinQuant + GQuant (Ours) | W3A4KV4 | **6.29** |

The results in Table 15 show that our method remains superior, though the gap narrows at larger bit-widths. We hypothesize that existing PTQ methods, which rely on less accurate surrogate objectives, have smaller gaps at higher bit-widths, allowing fine-tuning to narrow the difference. However, in more extreme compression settings, where the gap is wider, our method maintains its advantage even after fine-tuning.

### E.8. Results on Smaller Bit-width in Weight-and-activation Quantization

In weight-and-activation quantization, we further conduct an additional experiments with lower bit-widths for weights, specifically 2-bit and 3-bit, while keeping activations and KV caches at 4-bit precision (denoted as W2A4KV4 and W3A4KV4, respectively), on Llama-2-7B model. The results, shown in Table 16, demonstrate that GuidedQuant outperforms baseline methods by larger margin in these more extreme scenarios, highlighting the strength of our approach under stricter bit-width constraints.

### E.9. Comparison with mixed-precision variant of SqueezeLLM

The dense-and-sparse variant of SqueezeLLM (Kim et al., 2024), which preserves a small fraction of weights in 16-bit precision to maintain accuracy, is orthogonal to our method and can be combined with it. Accordingly, in Table 17, we report results for SqueezeLLM, LNQ, and LNQ + GuidedQuant methods, with the dense-and-sparse approach applied to all of them, using the identical experimental setting with Table 3. Following the original SqueezeLLM paper, we retain 0.45% of the weights in 16-bit and evaluate with 2-, 3-, and 4-bit quantization on the Llama-2-7B model. The results show that LNQ with GuidedQuant consistently outperforms the baselines in the dense-and-sparse setting as well, demonstrating the superiority and robustness of our method.

### E.10. Results on Different QTIP Variants (1MAD, 3INST, HYB)

The original QTIP paper introduced three variants of their method: 1MAD, 3INST, and HYB (Tseng et al., 2024b). Both 1MAD and 3INST are look-up table-free methods, while HYB incorporates a small look-up table that fits within the L1 cache of modern GPUs. The authors reported post-training quantization results without fine-tuning for the 1MAD and 3INST formats, while quantization with fine-tuning was reported for the HYB format. To maintain consistency, we report the better-performing variant between 1MAD and 3INST in Table 4 for both QTIP and our method (QTIP + GuidedQuant).

*Table 17.* Weight-only scalar post-training quantization results on Llama-2-7B model, evaluated under a dense-and-sparse setting, preserving 0.45% of the weights in 16 bits. Wiki2 and C4 denotes perplexity on WikiText2 and C4, respectively. The perplexity is measured with the context size of 4096.

| Method | Bits↓ | Wiki2↓ | C4↓ |
|---|---|---|---|
| Original | 16 | 5.12 | 6.63 |
| SqueezeLLM (0.45%) | 2.22 | 10.64 | 14.10 |
| LNQ (0.45%) (Ours) | 2.22 | 8.26 | 10.34 |
| LNQ + GuidedQuant (0.45%) (Ours) | 2.22 | **8.00** | **10.18** |
| SqueezeLLM (0.45%) | 3.24 | 5.58 | 7.23 |
| LNQ (0.45%) (Ours) | 3.24 | 5.49 | 7.15 |
| LNQ + GuidedQuant (0.45%) (Ours) | 3.24 | **5.48** | **7.12** |
| SqueezeLLM (0.45%) | 4.27 | 5.22 | 6.75 |
| LNQ (0.45%) (Ours) | 4.27 | **5.20** | 6.74 |
| LNQ + GuidedQuant (0.45%) (Ours) | 4.27 | **5.20** | **6.73** |

For completeness, the full performance results across 1MAD and 3INST are provided in Table 18.

It is worth noting that QTIP has only open-sourced the CUDA acceleration kernel for HYB, although it is theoretically possible to implement kernels for 1MAD and 3INST. Therefore, we also include the post-training quantization results (without fine-tuning) for the HYB format as well, summarized in Table 18. The results show that the variations among QTIP methods have minimal impact on the results, and our method consistently outperforms all others in Table 4, regardless of the QTIP variant chosen.

### E.11. Discussion on Block-diagonal Fisher Approximation

In this section, we review existing neural network compression methods that use a block-diagonal Fisher matrix approximation of the Hessian and highlight their differences from GuidedQuant. In particular, we discuss WoodFisher (Singh & Alistarh, 2020) for pruning CNNs, Optimal BERT Surgeon (Kurtic et al., 2022) for pruning BERT models, and BRECQ (Li et al., 2021) for quantizing CNNs.

WoodFisher and Optimal BERT Surgeon use blocks of arbitrary size $B \times B$ along the diagonal to reduce the storage cost. WoodFisher explores $B$ size of $\{20, 100, 1000, 5000, 12288, 37000\}$ in ResNet-20 (He et al., 2015), while Optimal BERT Surgeon uses $B = 50$, since the larger block size does not fit in the memory. BRECQ leaves the blocks that correspond to the parameters within each residual block in CNNs, and further uses a first-order Taylor approximation on the residual block's outputs to estimate the second-order error for each block to avoid the need to handle prohibitively large matrices.

The proposed GuidedQuant maintains the blocks corresponding to each output channel, resulting the $B$ size to be 4096 to 11008 for Llama-2-7B model. Directly computing these block-diagonal matrices would be infeasible, requiring over 110 TB for and more than $13,000$ GPU hours on RTX 6000 Ada GPU for Llama-2-7B. To address this, GuidedQuant averages the Fisher diagonal blocks within each group, approximately preserving dependencies within each output channel at the scale of modern LLMs. We present the theoretical complexity of GuidedQuant in Section 3.2, report its practical cost in Table 9, and report the performance of approximating more (opting for smaller number of groups) in Appendix E.5.

In Figures 3 and 4, we illustrate submatrices of the scaled Fisher information matrix, $n\mathbf{F}_j^{(l)} \times 10^6$, for the linear layers in the first Transformer block of the Llama-2-7B model, alongside corresponding approximation results. Here, $n$ denotes the number of calibration data, and the results are computed using calibration data from the RedPajama dataset, which consists of 1024 sentences with 4096 tokens each. Since each linear layer in the model contains $d_{in} \times d_{out}$ weights, fully visualizing its Fisher information matrix would yield a matrix of size $d_{in}d_{out} \times d_{in}d_{out}$, which is computationally prohibitive. Therefore, we restrict our visualization to the submatrix corresponding to the first two output channels of each layer. Since each output channel has $d_{in}$ weights, this results in visualizing a $2d_{in} \times 2d_{in}$ matrix. Within the Transformer block of the Llama-2-7B model, there are seven linear layers: self_attn.q_proj, self_attn.k_proj, self_attn.v_proj, self_attn.o_proj, mlp.gate_proj, mlp.up_proj, and mlp.down_proj. For the first six layers, $d_{in} = 4096$, so we visualize an $8192 \times 8192$ matrix, while for the final layer (mlp.down_proj) with $d_{in} = 11008$, an $22016 \times 22016$

*Table 18.* Weight-only post-training quantization results on different QTIP variants (1MAD, 3INST, HYB), *without fine-tuning* to the end-to-end loss. Wiki2 and C4 denotes perplexity on WikiText2 and C4, respectively, which are measured with the context size of 4096.

| Variant | Method | Bits↓ | Llama-2-7B | | Llama-2-13B | | Llama-2-70B | |
|---|---|---|---|---|---|---|---|---|
| | | | Wiki2↓ | C4↓ | Wiki2↓ | C4↓ | Wiki2↓ | C4↓ |
| | Original | 16 | 5.12 | 6.63 | 4.57 | 6.05 | 3.12 | 4.97 |
| 1MAD | QTIP | 2.00 | 7.05 | 9.14 | 5.59 | 7.46 | 3.87 | 5.70 |
| | QTIP + GQuant (Ours) | 2.00 | **6.11** | **7.99** | **5.33** | **7.05** | **3.80** | **5.61** |
| | QTIP | 3.00 | 5.38 | 6.99 | 4.74 | 6.28 | 3.27 | 5.09 |
| | QTIP + GQuant (Ours) | 3.00 | **5.28** | **6.87** | **4.71** | **6.22** | **3.25** | **5.08** |
| | QTIP | 4.00 | 5.17 | 6.71 | 4.62 | 6.10 | 3.16 | **5.00** |
| | QTIP + GQuant (Ours) | 4.00 | **5.16** | **6.68** | **4.61** | **6.09** | **3.15** | **5.00** |
| 3INST | QTIP | 2.00 | 6.82 | 8.96 | 5.52 | 7.39 | 3.90 | 5.69 |
| | QTIP + GQuant (Ours) | 2.00 | **6.16** | **7.99** | **5.33** | **7.04** | **3.82** | **5.61** |
| | QTIP | 3.00 | 5.40 | 7.01 | 4.74 | 6.28 | 3.27 | 5.09 |
| | QTIP + GQuant (Ours) | 3.00 | **5.30** | **6.87** | **4.70** | **6.22** | **3.26** | **5.08** |
| | QTIP | 4.00 | 5.17 | 6.71 | 4.62 | 6.10 | 3.16 | **5.00** |
| | QTIP + GQuant (Ours) | 4.00 | **5.16** | **6.68** | **4.61** | **6.09** | **3.15** | **5.00** |
| HYB | QTIP | 2.00 | 6.84 | 9.03 | 5.62 | 7.46 | 3.93 | 5.74 |
| | QTIP + GQuant (Ours) | 2.00 | **6.19** | **8.06** | **5.36** | **7.10** | **3.84** | **5.64** |
| | QTIP | 3.00 | 5.39 | 7.03 | 4.76 | 6.31 | 3.28 | 5.10 |
| | QTIP + GQuant (Ours) | 3.00 | **5.32** | **6.89** | **4.72** | **6.24** | **3.27** | **5.09** |
| | QTIP | 4.00 | 5.19 | 6.73 | 4.63 | 6.12 | 3.17 | 5.01 |
| | QTIP + GQuant (Ours) | 4.00 | **5.18** | **6.70** | **4.61** | **6.10** | **3.16** | **5.00** |

matrix is visualized.

We compare two approximation strategies:

- WoodFisher: This approach retains the blocks size of $B \times B$ along the diagonal. The storage requirement for this method is $B \, d_{\text{in}} \, d_{\text{out}}$.

- GuidedQuant: Here, the block size is set to $d_{\text{in}} \times d_{\text{in}}$ and blocks are averaged within groups. This strategy requires $g \, d_{\text{in}}^2$ storage, where $g$ is the number of groups.

To ensure a fair comparison, we choose the WoodFisher block size as $B = \lceil g \, d_{\text{out}}/d_{\text{in}} \rceil$. Specifically, we choose $g = 4$ for the GuidedQuant, which results in $B = 4$ for the self-attention projection layers, $B = 2$ for the `mlp.gate_proj` and `mlp.up_proj` layers, and $B = 11$ for the `mlp.down_proj` layer.

The visualizations reveal that the original Fisher information matrix exhibits strong off-diagonal values and a prominent block-diagonal structure with blocks of size $d_{\text{in}} \times d_{\text{in}}$. This indicates stronger interactions among weights within the same output channel compared to those across different channels. Overall, the GuidedQuant approximation captures significantly more of this structural detail than the WoodFisher-style block-diagonal approximation, which retains only arbitrarily sized diagonal blocks.

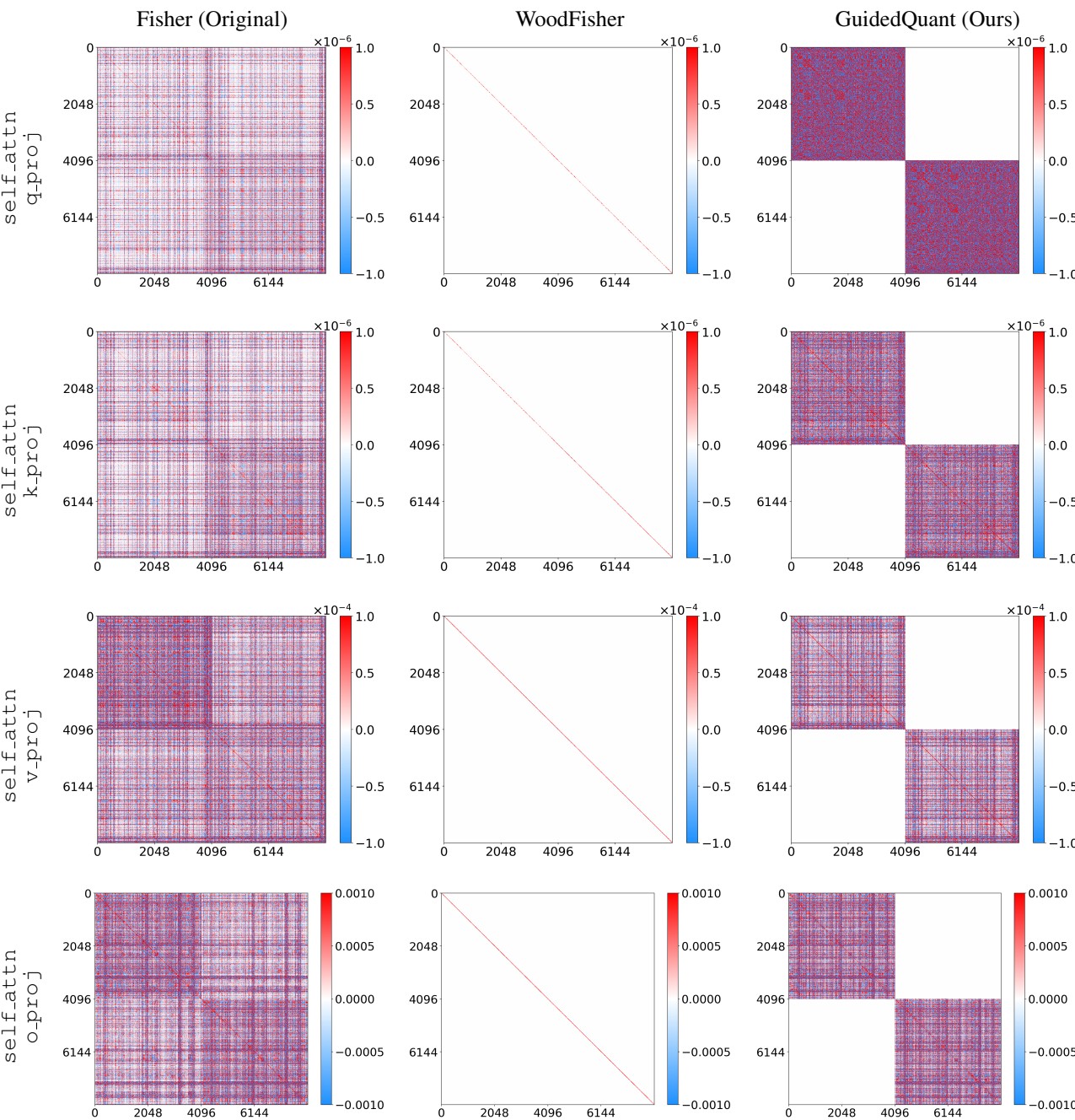

*Figure 3.* Visualization of the scaled Fisher information matrix, $n\mathbf{F}_j^{(l)} \times 10^6$, for the first two output channels in the self_attn.q_proj, self_attn.k_proj, self_attn.v_proj, and self_attn.o_proj layer of the first Transformer block in Llama-2-7B model. Left: the original Fisher matrices; Middle: the WoodFisher style block-diagonal approximation (block size $B = 4$ for all of the layers); Right: the GuidedQuant approximation (the number of groups $g = 4$). Both approximations are compared under an equal storage budget.

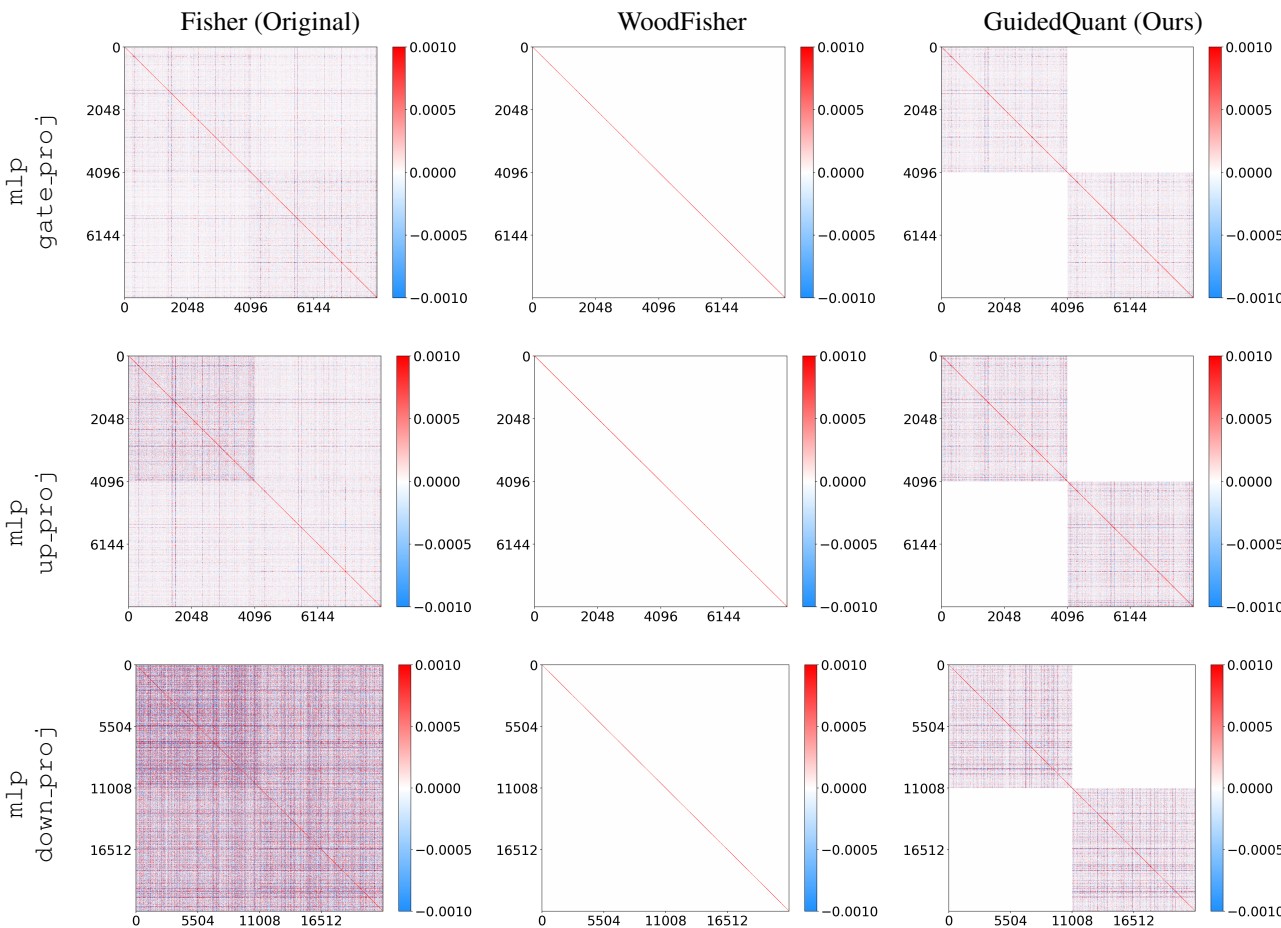

*Figure 4.* Visualization of the scaled Fisher information matrix, $n\mathbf{F}_j^{(l)} \times 10^6$, for the first two output channels in the mlp.gate_proj, mlp.up_proj, and mlp.down_proj layer of the first Transformer block in Llama-2-7B model. Left: the original Fisher matrices; Middle: the WoodFisher style block-diagonal approximation (block size $B = 2$, $B = 2$, and $B = 11$, respectively); Right: the GuidedQuant approximation (the number of groups $g = 4$). Both approximations are compared under an equal storage budget.

