# OpenReview forum: "GuidedQuant: Large Language Model Quantization via Exploiting End Loss Guidance"
_ICML.cc/2025/Conference — ICML 2025 poster_

### Official Review · Reviewer_UEhr · 2025-03-09

**Overall Recommendation:** 3

**Summary:**

This paper introduces GuidedQuant, a novel quantization framework that integrates gradient information from the end-to-end loss into the quantization objective while explicitly modeling inter-weight dependencies. The authors first identify a critical limitation of existing quantization methods: they either treat all hidden features equally or, when incorporating end-to-end loss, fail to capture the cross-interaction between weights.

To address these challenges, the paper proposes guided quantization, which leverages gradient information from the end-to-end loss to reweight layer-wise quantization errors adaptively. Furthermore, the authors introduce Layerwise Non-Uniform Quantization (LNQ), a novel non-uniform scalar quantization method that employs a closed-form solution for optimizing the codebook and a coordinate descent algorithm for optimizing the assignment matrix.

**Claims And Evidence:**

- The authors theoretically derive an objective function that accounts for the importance of each layer in Section 3.1.
- They provide a proof in Appendix A to support Proposition 3.1, which states that minimizing their proposed objective function is equivalent to minimizing a second-order error estimate incorporating the Fisher information matrix, including inter-weight interaction terms.

- Their claims are also supported by evaluation results presented in Section 5 and Appendices C and D.

**Essential References Not Discussed:**

None

**Experimental Designs Or Analyses:**

- The experimental designs are well established to support the paper's main claim and address potential concerns about the proposed methods' weaknesses.

- However, authors mentioned they used perplexity values reported in previous papers for several baselines. However, perplexity can vary depending on evaluation setups and is rarely reproducible across different setups. If the authors aim to claim state-of-the-art performance, they should evaluate baseline methods in the same environment to ensure a fair comparison.

- Moreover, while the proposed methods achieve the best results, the accuracy improvement appears marginal, particularly in 3-bit and 4-bit quantization. Broader evaluations across diverse architectures and tasks would strengthen empirical validation.

**Methods And Evaluation Criteria:**

The proposed methods are well-justified for the following reasons.
- They address the limitations of existing quantization methods, which fail to account for the varying importance of layer-wise outputs in relation to end-to-end loss.
- They also mitigate another drawback of end-to-end loss-based approaches, which tend to overlook inter-weight dependencies.
- The guided quantization framework is adaptable and can be integrated with various existing quantization methods.

However, the proposed methods lack novelty.
- The objective function is derived from changes in the end-to-end loss, which is similar to previous work.
- The proposed approach to addressing increased computational and memory costs is relatively straightforward. More ablation studies are needed to either substantiate the claim that a simple method is sufficient or to explore ways to mitigate accuracy degradation in 2-bit quantization.

Additionally, evaluation criteria is somewhat limited.
- The results are presented only on LLaMA 2 models using perplexity on the Wikitext-2 and C4 datasets. To better support the effectiveness of the proposed methods, evaluations on additional benchmark datasets (e.g., MMLU, zero-shot evaluations) should be included.
- Since uniform scalar quantization is widely used, further comparisons with this approach would provide a more comprehensive assessment of the proposed method’s effectiveness.

**Other Comments Or Suggestions:**

- Can you clarify how GPU cost in Table7 and Table8  is measured?

- In Section 5.1, you mentioned applying GuidedQuant to non-uniform scalar quantization and vector quantization. However, Table 3 includes GPTQ, which uses uniform scalar quantization. If you applied a non-uniform scalar quantization method for the GPTQ baseline, could you provide a more detailed explanation? If not, further clarification on this would be helpful.

**Other Strengths And Weaknesses:**

I have discussed the strengths and weaknesses of this paper in the preceding sections.

**Questions For Authors:**

Comments and suggestions section covers up my questions.

**Relation To Broader Scientific Literature:**

The key contributions of this paper relate to the literature on LLM post-training quantization (PTQ), which aims to minimize layer-wise reconstruction error. While adopting the concept of layer-wise error minimization, this paper introduces a new objective function that accounts for both layer-wise importance and inter-weight interactions.

**Theoretical Claims:**

In Section 4.2, the closed-form solution for optimizing the codebook is theoretically well-motivated; however, it lacks a formal proof or derivation.

---

> ### Author Rebuttal · Authors · 2025-03-31
>
> Thank you for the valuable feedback. We address below your questions and concerns.
>
> ---
>
> **Q1. The objective is derived from changes in the end-to-end loss, which is similar to previous work & Concern on the simplicity of the approximation method.**
>
> We believe the simplicity of our approach should not be mistaken for a lack of novelty. On the contrary, we view the simplicity of our approach as a strength rather than a limitation, as it achieves strong and robust performance gains through a simple and efficient method (see Figure 2 and Tables 3-5 of the paper).
>
> That said, we agree that exploring alternative grouping strategies, such as incorporating sophisticated clustering algorithms for more accurate Hessian approximation, is a promising direction for future research.
>
> ---
>
> **Q2. Evaluations on zero-shot and few-shot downstream tasks**
>
> Please refer to **our response to Q1 from Reviewer WqtU**, where we evaluate our method alongside baselines on eight zero-shot downstream tasks and the MMLU benchmark under a 5-shot setting. The results show that LNQ combined with GuidedQuant consistently matches or surpasses baseline performance, with especially notable improvements under extreme quantization scenarios.
>
> ---
>
> **Q3. Comparison with uniform scalar quantization methods**
>
> We note that comparisons with uniform scalar quantization methods are included in our experiments. Specifically, Table 3 includes the baselines of weight-only uniform scalar quantization methods such as GPTQ and QuIP, and Table 5 includes comparisons for weight-and-activation uniform scalar quantization methods like QuaRot and SpinQuant. Our method consistently outperforms these baseline approaches in both weight-only and weight-and-activation settings, and shows effectiveness even when used with a uniform quantization scheme as well (SpinQuant + GuidedQuant (Ours) in Table 5).
>
> ---
>
> **Q4. Evaluations on different models**
>
> Please refer to **our response to Q3 from Reviewer NTd9**, where we conduct more experiments on Llama-3-8B and Llama-3-70B under a weight-only scalar quantization setting. The results show that our method consistently outperforms the baseline methods in these settings as well.
>
> ---
>
> **Q5. Derivation for closed-form solution in Section 4.2**
>
> The closed-form solution $\mathbf{c}^\ast$ in Section 4.2 (Eq. 10) is the standard least squares solution
>  $\mathbf{c = A^\dagger y}$ for the problem $\min_{\mathbf{c}} \\| \mathbf{y - A c} \\|^2$, where $\mathbf{y = X w}$ and $ \mathbf{A =X P}$.  We assume that $\bf{A^\top A}$ is invertible, hence $\mathbf{A^\dagger = (A^\top A)^{-1} A^\top}$. This is a common assumption in quantization.
> In practice, $\bf{A^\top A}$ is not always invertible. To address this, we add $\bf \lambda I$ to $\bf{A^\top A}$, with a small $\lambda = 10^{-7}$, as commonly done in prior work.
>
> ---
>
> **Q6. Clarification on the perplexity evaluation setups and ensuring a fair comparison**
>
> To ensure a fair comparison, we use the perplexity values reported in previous papers only when the evaluation setup strictly matches ours, i.e., same calibration and evaluation dataset, context length, and a calibration data size that is equal to or larger than ours (giving the baselines an advantage). When any of these criteria were not met, we reproduced the baseline methods under the exact same setup as our experiments. We will clarify this in the revision.
>
> ---
>
> **Q7. Clarification on measuring GPU cost in Table 7 and 8**
>
> In Tables 7 and 8, we report the end-to-end runtime for each quantization method, along with the number and type of GPUs used (indicated in parentheses). For example, "1×R6A" refers to the runtime measured using a single RTX 6000 Ada GPU, while "4×R6A" indicates that the method was run using four RTX 6000 Ada GPUs.
>
> ---
>
> **Q8. Marginal improvements in quantization results with higher bits**
>
> The performance gains from our method are especially prominent in more challenging quantization scenarios. In particular, our approach demonstrates clearer advantages in lower-bit settings or when applied to models that are inherently harder to quantize, such as Llama-3 models. In these settings, our method yields more substantial improvements. We will include these discussions and corresponding results in the revision.
>
> ---
>
> **Q9. Non-uniform scalar quantization format for the GPTQ method**
>
> Non-uniform scalar quantization maps weights to non-uniform grids. GPTQ is designed to minimize the layer-wise output reconstruction error for a fixed quantization grid, thus it can be extended to non-uniform grids by also optimizing the choice of the grid.
> The GPTVQ 1D baseline does this; it applies the GPTQ algorithm to optimize assignments for a fixed quantization grid while optimizing the non-uniform codebook using gradient descent. We included GPTVQ 1D as a baseline in Table 3 in the paper and showed that both LNQ and LNQ with GuidedQuant consistently outperform it across all evaluated models and bit-widths.

---

### Official Review · Reviewer_i4yb · 2025-03-10

**Overall Recommendation:** 4

**Summary:**

The paper introduces GuidedQuant, a layer-wise quantization method that also considers its impact on the final loss. Through derivation, the authors propose a new objective that reweights the Hessian using the loss with respect to the layer outputs. To mitigate computational costs, they approximate the Hessian by averaging it over groups. Additionally, the paper presents an EM-based algorithm (LNQ) to learn quantization scalars.

The proposed methods are evaluated on LLaMA2 across various quantization settings, where GuidedQuant consistently outperforms existing approaches in nearly all scenarios.

**Update after rebuttal**: My latest reply reflected my final update.

**Claims And Evidence:**

Yes.

**Essential References Not Discussed:**

No.

**Experimental Designs Or Analyses:**

Yes.

However, I have question on how is the caching is calculated in table 8. For gradient respect to outputs, I thought the approximate cost for a LLaMA2-7B model is
1024 (batch size) * 4096 (sequence length) * 4000 (dimension) * 32 (number of layers) * 7 (number of modules per layers) * 2 (if store in 16bits) / 10 ** 9 = 7516 GB, the same question also extends to Hessian caching.

**Methods And Evaluation Criteria:**

Yes. However, it would be great to evaluate on newer models and on some downstream tasks, as perplexity sometimes doesn't reflect the downstream performance.

**Other Comments Or Suggestions:**

N/A

**Other Strengths And Weaknesses:**

[Strength]

* The paper introduces an useful improvement to equip layer-wise quantization with awareness of final output loss.
* The paper is well-written and easy to follow.
* The experimental results are strong and generalize to different quantization methods.

[Weakness]

I only have some minor comments which can be found in "Methods And Evaluation Criteria" and "Experimental Designs Or Analyses".

**Questions For Authors:**

See Other Strengths And Weaknesses.

**Relation To Broader Scientific Literature:**

The findings can be applied to layer-wise pruning as well.

**Theoretical Claims:**

Yes. I checked the proof of proposition 3.1 and I think there are no issues.

---

> ### Author Rebuttal · Authors · 2025-03-31
>
> Thank you for your positive review. We address below your questions.
>
> ---
>
> **Q1. Results on newer models**
>
> Please refer to **our response to Q3 from Reviewer NTd9**, where we conduct more experiments on Llama-3-8B and Llama-3-70B under a weight-only scalar quantization setting. The results show that our method consistently outperforms the baseline methods in these settings as well.
>
> ---
>
> **Q2. Evaluations on zero-shot and few-shot downstream tasks**
>
> Please refer to **our response to Q1 from Reviewer WqtU**, where we evaluate our method alongside baselines on eight zero-shot downstream tasks and the MMLU benchmark under a 5-shot setting. The results show that LNQ combined with GuidedQuant consistently matches or surpasses baseline performance, with especially notable improvements under extreme quantization scenarios.
>
> ---
>
> **Q3. Clarification on the gradient / Hessian cache size**
>
> We note that GuidedQuant only requires the Hessian averaged within groups for each layer. Since this averaged Hessian can be computed using the group-wise averaged gradient values, there is no need to cache the full gradient with respect to the output tensor. Instead, we only store one averaged scalar gradient value per group, per layer.
>
> For example, in the case of the Llama-2-7B model with $g=4$ number of groups, each layer only needs to cache 4 scalar values per layer for averaged gradients.
> The total gradient cache size is therefore computed as:
>
> - $1024$ (batch size) $\times$ $4096$ (sequence length) $\times$ $4$ (number of groups) $\times$ $32$ (number of layers) $\times$ $7$ (number of modules per layers) $\times$ $2$ bytes (stored in 16-bits) = $7.5 \text{GB}$.
>
> This averaging based approximation also applies to the Hessian computation, substantially reducing the storage cost compared to storing the full Hessian without approximation. This is how the cache sizes reported in Table 8 in the paper are derived. We will clarify this in the revision.

---

> > ### Comment · Reviewer_i4yb · 2025-04-05
> >
> > Thanks to the authors for the rebuttal contents. My concerns are addressed and I will keep my rating as "accept".

---

> > > ### Author Response · Authors · 2025-04-06
> > >
> > > Dear Reviewer i4yb,
> > >
> > > Thank you for your response and for your positive review. We truly appreciate the time and effort you have spent providing valuable feedback.
> > >
> > > Best regards, Authors.

---

### Official Review · Reviewer_NTd9 · 2025-03-13

**Overall Recommendation:** 4

**Summary:**

This paper introduces GuidedQuant, a post-training quantization framework for large language models (LLMs) that integrates gradient information from the end-to-end loss into the quantization objective while explicitly modeling inter-weight dependencies. The authors claim that GuidedQuant improves the performance of state-of-the-art quantization algorithms across scalar and vector quantization methods. Additionally, they propose a non-uniform scalar quantization algorithm, LNQ, which outperforms existing methods in this category. The paper presents experimental results on Llama-2 models.

**Claims And Evidence:**

- GuidedQuant “explicitly models pairwise interactions between weights.” **Issue**: While Proposition 3.1 links the objective to second-order error estimates, the actual implementation approximates Hessian matrices via grouping (Section 3.3), which dilutes the theoretical benefit. No empirical evidence shows improved modeling of interactions.
- LNQ “outperforms existing non-uniform scalar quantization methods.” **Issue**: Comparisons to GPTVQ 1D and SqueezeLLM are provided, but SqueezeLLM’s mixed-precision variant (Kim et al., 2024) is excluded.

**Essential References Not Discussed:**

Not applicable.

**Experimental Designs Or Analyses:**

Table 10 (Appendix D.1) shows minimal gains with larger groups, suggesting the method is not robust to hyperparameters.

**Methods And Evaluation Criteria:**

- The grouping heuristic is reminiscent of block-wise quantization (Dettmers et al., 2022) but lacks justification.
- Results are restricted to Llama-2 models. No experiments on other architectures.

**Other Comments Or Suggestions:**

**Typos**:
   - Page 3: \(\mathbf{z}_{j}\in\mathbb{R}^{d_{\text{out}}}\) has a mismatched bracket.
   - Table 1: “WAA4KV4” is undefined.

**Repetition**: Section 2 redundantly explains basics of quantization.
**Clarity**: Figure 1’s caption is overly technical.

**Other Strengths And Weaknesses:**

Not applicable.

**Questions For Authors:**

No.

**Relation To Broader Scientific Literature:**

GuidedQuant builds on second-order optimization (Hassibi & Stork, 1992) and layer-wise reconstruction (Frantar et al., 2022).

**Theoretical Claims:**

- LNQ’s closed-form codebook update (Equation 10) is a theoretically sound improvement over gradient-based fine-tuning.

- The proof (Appendix A) correctly links the objective to the Fisher matrix but assumes gradients are computed precisely. In practice, gradients are noisy due to calibration data  (Frantar et al., 2022), which is unaddressed.

---

> ### Author Rebuttal · Authors · 2025-03-31
>
> Thank you for your positive review. We address below your questions.
>
> ---
>
> **Q1. Lack of empirical evidence showing improved modeling after grouping approximation (Section 3.3).**
>
> We highlight that the GuidedQuant objective, after applying the grouping approximation, consistently demonstrates a clear empirical advantage over both the layer-wise objective and the diagonal Hessian approximation (i.e., the weighted $k$-means objective). This is evidenced by the performance gains shown in Figure 2, Tables 3, 4 and 5 of the paper.
>
> To provide further qualitative evidence, we visualize a submatrix of the Fisher information matrix corresponding to the first two output channels in the linear layers of the first Transformer block of Llama-2-7B in *Figures A1 and A2*. Since each output channel contains  $d_{\mathrm{in}}$  weights, the visualized matrix has dimensions of $2 d_{\mathrm{in}} \times 2 d_{\mathrm{in}}$ (here, $d_{\mathrm{in}}=4096$ for all layers except `mlp.up_proj` for which  $d_{\mathrm{in}}=11008$).
> The visualizations reveal that the original Fisher information matrix exhibits strong off-diagonal values and a prominent block-diagonal structure with blocks of size $d_{\mathrm{in}} \times d_{\mathrm{in}}$, corresponding to interactions within each of the two output channels. GuidedQuant approximately preserves the off-diagonal terms within each output channel, thereby capturing significantly more structural detail than the diagonal approximation (SqueezeLLM), which ignores the cross-weight interactions. We believe this better structural modeling helps explain the performance gains achieved by our method. We will include these results and discussions in the revision.
>
> - *Figure A1: https://drive.google.com/file/d/1ilhJarLN1u1nNBzT2ywDCxg8IRz-T2oz/view?usp=sharing*
> - *Figure A2: https://drive.google.com/file/d/19LJkHuKebYyOvPsqsjJQG_5PkJAWr7Bz/view?usp=sharing*
>
> ---
>
> **Q2. Comparison with mixed-precision variant (dense-and-sparse) of SqueezeLLM.**
>
> Please refer to **our response to Q2 from Reviewer WqtU**, where we present a comparison between our method and SqueezeLLM under a mixed-precision (dense-and-sparse) setup. The results show that our method consistently outperforms the SqueezeLLM in this setting as well.
>
> ---
>
> **Q3. Results on newer models other than Llama-2 (Reviewers NTd9, i4yb, UEhr)**
>
> We have conducted additional experiments on newer models, including Llama-3-8B and Llama-3-70B, comparing our method with SqueezeLLM under a weight-only scalar quantization setting. We present the results in *Table A4*. LNQ with GuidedQuant consistently outperforms the baselines, demonstrating the robustness and effectiveness of our approach. We plan to conduct more comprehensive experiments on these newer models and will include the results in the revision.
>
> - *Table A4: https://drive.google.com/file/d/11bgdJ5eOO5s3LjjcZgVPyAzPsXDr6JJZ/view?usp=sharing*
>
> ---
>
> **Q4. The gradients are computed on calibration data, which is unaddressed in Appendix A when connecting the objective to the Fisher matrix.**
>
> In Proposition 3.1, we define the Fisher information matrix $\mathbf{F}$ as the *empirical* Fisher information matrix computed over the calibration data, $\mathbf{F} = \frac{1}{n} \sum_{i=1}^n \nabla \ell_i (\mathbf{w}) \nabla \ell_i (\mathbf{w})^\top$. Accordingly, the proof assumes that gradients are computed using the calibration data and is connecting the objective to this empirical Fisher information matrix. In the revision, we will explicitly refer to $\mathbf{F}$ as the empirical Fisher information matrix on the calibration data and clarify that it is an approximation of the true Fisher information matrix.
>
> ---
>
> **Q5. Minimal gains with a bigger number of groups, and lack of explanation on the choice of hyperparameter $g$ (e.g., $g=4$ in 7B models).**
>
> We believe that achieving strong performance even with a small number of groups is a strength of our method, making it particularly effective and robust in resource-constrained settings. Regarding the hyperparameter choice, we selected the number of groups $g$ to be as large as possible within the limits of our computational and memory constraints. We will clarify this in the revision.
>
> ---
>
> **Q6. Additional comments and suggestions.**
>
> Thank you for the suggestions. We will revise the writing to improve overall clarity, including correcting typos, reducing redundant explanations, and refining figure captions for clarity.

---

> > ### Comment · Reviewer_NTd9 · 2025-04-03
> >
> > I appreciate the authors’ response in the rebuttal and my concerns are addressed. Therefore, I incline to raise my rating.

---

> > > ### Author Response · Authors · 2025-04-03
> > >
> > > Dear Reviewer NTd9,
> > >
> > > Thank you for your response and for increasing your score. We truly appreciate the time and effort you have spent providing valuable feedback.
> > >
> > > Best regards,
> > > Authors.

---

### Official Review · Reviewer_WqtU · 2025-03-14

**Overall Recommendation:** 3

**Summary:**

This paper proposes to use layer output combined with gradient information as the objective to minimize layer-wise quantization perturbation, along with an approximate method to solve this resource-intensive problem. By improving GPTVQ and combining it with the new objective proposed in the paper, the experiment performed well on non-uniform quantization and also showed some improvements in uniform quantization.

**Claims And Evidence:**

Yes, it is clear. The method proposed is relatively clear and simple, and the paper uses sufficient empirical study to prove the effectiveness of the proposed method.

**Essential References Not Discussed:**

No.

**Experimental Designs Or Analyses:**

Yes, I've checked the main experiment, including the weight-only PTQ and weight-activation PTQ parts. My main concern comes from the fact that the evaluation is limited to model perplexity.

**Methods And Evaluation Criteria:**

Not sufficient. As an LLM PTQ paper, the article only involves model perplexity as the only evaluation indicator, and does not test it on any generative tasks or other few-shot benchmarks, which is inconsistent with convention.

**Other Comments Or Suggestions:**

1. The third entry in Table8 should be "Llama-2-70B" instead of "Llama-2-7B"

**Other Strengths And Weaknesses:**

# Strengths:
1. The paper is well written, and the previous knowledge is fully explained and introduced, making it easy for readers to follow.
2. The results on non-uniform quantization show significant improvement over existing methods
# Weaknesses:
1. In the weight-activation quantization experiment(i.e. table 5), it seems to show that GuidedQuant does not show obvious advantage than GPTQ
2. It is recommended that the authors supplement the test results of generative tasks and other commonly used zero-shot tasks, such as MMLU.

**Questions For Authors:**

1. Why dense-sparse separated quantization is not enabled for squeezellm ? This setting may significantly improve the results according to the results reported in the original paper.

**Relation To Broader Scientific Literature:**

The key contribution of this paper can be viewed as an improvement of previous layer-wise error reconstruction problems. By Introducing gradient information as an importance indicator, the reconstruction process can be more accurate and the empirical study shows better results.

**Theoretical Claims:**

Yes, I've checked the theoretical analysis in Section 3.1.

---

> ### Author Rebuttal · Authors · 2025-03-31
>
> Thank you for your positive review. We address below your questions.
>
> ---
> **Q1. Evaluations on zero-shot and few-shot downstream benchmarks (Reviewers WqtU, i4yb, UEhr).**
>
> We provide evaluations of our methods (LNQ and LNQ + GuidedQuant) alongside baselines (SqueezeLLM and GPTVQ 1D) under the weight-only scalar quantization settings, using Llama-2-7B and Llama-2-13B models, in *Table A1*. The evaluation includes eight zero-shot tasks: BoolQ [1], PIQA [2], SIQA [3], HellaSwag [4], WinoGrande [5], ARC-easy [6], ARC-challenge [6], and OBQA [7]. For a few-shot benchmark, we include results on the MMLU [8] benchmark in a 5-shot setting.
>
> *Table A1* reports both accuracy and standard error for all methods. We highlight the best-performing results, as well as those whose accuracy falls within the top score $\pm$ standard error, under the same bit width constraint. The results show that LNQ combined with GuidedQuant consistently matches or surpasses baseline performance, with notable improvements in extreme quantization scenarios, such as 2-bit quantization. While we currently report results on a subset of models and settings due to time constraints, we will perform a more comprehensive set of experiments and include the full results in the revision.
>
> - *Table A1: https://drive.google.com/file/d/1XnAs0V8CwTKpoul5I3cIVecE5JL8cgto/view?usp=sharing*
>
>
> ---
>
> **Q2. Comparison with Dense-and-Sparse variant of SqueezeLLM (Reviewers WqtU, NTd9).**
>
> To ensure a fair comparison, all methods in the paper were evaluated under uniform precision. The dense-and-sparse variant of SqueezeLLM, which preserves a small fraction of weights in 16-bit precision to maintain accuracy, is orthogonal to our method and can be combined with it. Accordingly, we report results for SqueezeLLM, LNQ, and LNQ + GuidedQuant methods, with the dense-and-sparse approach applied to all of them, in *Table A2*.
>
> Following the original SqueezeLLM paper, we retain 0.45% of the weights in 16-bit and evaluate with 2-, 3-, and 4-bit quantization on the Llama-2-7B model. The results show that LNQ with GuidedQuant consistently outperforms the baselines in the dense-and-sparse setting as well, demonstrating the superiority and robustness of our method. We will include these results in the revision.
>
> - *Table A2: https://drive.google.com/file/d/1dORQnlmUi1FPQmuSR8n1OebNQ5h0M8Gi/view?usp=sharing*
>
> ---
>
> **Q3. In weight-and-activation quantization (Table 5 in the paper), the GuidedQuant does not show obvious advantage over the baseline.**
>
> This is primarily because we present weight-and-activation results using relatively high bit-width for weight quantization (4-bit), where the benefits of our method are less evident. Our method demonstrates more significant improvements in more aggressive quantization settings.
> To illustrate this, we conducted additional experiments with lower bit-widths for weights, specifically 2-bit and 3-bit, while keeping activations and KV caches at 4-bit precision (denoted as W2A4KV4 and W3A4KV4, respectively), on Llama-2-7B model. The results, shown in *Table A3* below, demonstrate that GuidedQuant significantly outperforms baseline methods in these more extreme scenarios, highlighting the strength of our approach under stricter bit-width constraints. We will include these new results and the corresponding discussion in the revision.
>
> - *Table A3: https://drive.google.com/file/d/1PaDjWnzsWEYVd5Lwdd_Xoath72Fwfghr/view?usp=sharing*
>
> ---
>
> **References**
>
>
> [1] Christopher Clark, Kenton Lee, Ming-Wei Chang, Tom Kwiatkowski, Michael Collins, and Kristina Toutanova. Boolq: Exploring the surprising difficulty of natural yes/no questions. 2019.
>
> [2] Yonatan Bisk, Rowan Zellers, Ronan Le Bras, Jianfeng Gao, and Yejin Choi. Piqa: Reasoning about physical commonsense in natural language. 2020.
>
> [3] Maarten Sap, Hannah Rashkin, Derek Chen, Ronan LeBras, and Yejin Choi. Socialiqa: Commonsense reasoning about social interactions. 2019.
>
> [4] Rowan Zellers, Ari Holtzman, Yonatan Bisk, Ali Farhadi, and Yejin Choi. Hellaswag: Can a machine really finish your sentence? 2019.
>
> [5] Keisuke Sakaguchi, Ronan Le Bras, Chandra Bhagavatula, and Yejin Choi. Winogrande: An adversarial winograd schema challenge at scale. 2021.
>
> [6] Peter Clark, Isaac Cowhey, Oren Etzioni, Tushar Khot, Ashish Sabharwal, Carissa Schoenick, and Oyvind Tafjord. Think you have solved question answering? try arc, the ai2 reasoning challenge. 2018.
>
> [7] Todor Mihaylov, Peter Clark, Tushar Khot, and Ashish Sabharwal. Can a suit of armor conduct electricity? a new dataset for open book question answering. 2018.
>
> [8] Hendrycks, D., Burns, C., Basart, S., Zou, A., Mazeika, M., Song, D. and Steinhardt, J., Measuring massive multitask language understanding. 2020.

---

### Decision · Program_Chairs · 2025-05-01

**Decision:**

Accept (poster)

**Comment:**

This paper introduces GuidedQuant, a novel post-training quantization framework that integrates gradient information from the end-to-end loss and explicitly models inter-weight dependencies. After the rebuttal, it receives two accept, one weak accept, and one weak reject. Its merits, including the interesting idea, good organization and writing, extensive experiments and good results, are well recognized by the reviewers. The response well addresses the reviewers' concerns about the additional evaluation and comparison, detailed analysis, and so on. I think the current manuscript meets the requirement of this top conference and recommend for acceptance. Please incorporate the revision in the updated manuscript.